# WASH maintains NKp46+ ILC3 cells by promoting AHR expression

Pengyan Xia[1,*], Jing Liu[1,2,*], Shuo Wang[1,*], Buqing Ye[1], Ying Du[1], Zhen Xiong[1,2], Ze-Guang Han[3], Liang Tong[4] & Zusen Fan[1,2]

Innate lymphoid cells (ILCs) communicate with other haematopoietic and non-haematopoietic cells to regulate immunity, inflammation and tissue homeostasis. How these ILC lineages develop and are maintained is not clear. Here we show that WASH is highly expressed in the nucleus of group 3 ILCs (ILC3s). WASH deletion impairs the cell pool of NKp46+ ILC3s. In NKp46+ ILC3s, WASH recruits Arid1a to the Ahr promoter thus activating AHR expression. WASH deletion in ILC3s decreases the number of NKp46+ ILC3s. Moreover, Arid1a deletion impedes AHR expression and impairs the maintenance of NKp46+ ILC3s. Therefore, WASH-mediated AHR expression has a critical function in the maintenance of NKp46+ ILC3s.

[1] Key Laboratory of Infection and Immunity of CAS, CAS Center for Excellence in Biomacromolecules, Institute of Biophysics, Chinese Academy of Sciences, Beijing 100101, China. [2] University of Chinese Academy of Sciences, Beijing 100049, China. [3] Key Laboratory of Systems Biomedicine (Ministry of Education) and Collaborative Innovation Center of Systems Biomedicine, Shanghai Center for Systems Biomedicine, Shanghai Jiao Tong University, Shanghai 200240, China. [4] Department of Biological Sciences, Columbia University, New York, New York 10027, USA. * These authors contributed equally to this work. Correspondence and requests for materials should be addressed to Z.F. (email: fanz@moon.ibp.ac.cn).

nnate lymphoid cells (ILCs) reside in mucosal surfaces to potentiate immune responses, sustain mucosal integrity and maintain tissue homeostasis. ILCs can be categorized into three groups based on their signature effector cytokines, analogous to the classification of T cell subsets[1]. Group 1 (ILC1) cells are characterized by their capacity to secrete interferon γ (IFN-γ) in response to interleukin 12 (IL-12), IL-15 and IL-18 (refs 2,3). Group 2 (ILC2) cells generate type 2 T helper (Th2) cell cytokines such as IL-5, IL-9 and IL-13 in response to IL-25 and IL-33 stimulation[4–6]. Group 3 (ILC3) cells produce IL-17 and IL-22 upon stimulation with IL-1β and IL-23 (refs 7–9). ILC3 cells can be divided into subpopulations by their expression of CD4 and NKp46 (encoded by Ncr1) receptors, such as CD4$^+$ ILC3s, NKp46$^+$ ILC3s and CD4$^-$NKp46$^-$ ILC3s (DN ILC3s)[1,10]. ILC3 subsets in the fetal intestines are CD4$^+$ or CD4$^-$ lymphoid tissue inducer (LTi) cells, which are necessary for the development of Peyer's patches (PPs) and lymph nodes[11]. In addition, ILC3s are central to the defence against bacterial infection in the intestine[12,13]; NKp46$^+$ ILC3s, for example, specifically generate IL-22 but not IL-17 (refs 14,15).

ILC3s are enriched in PPs and intestinal lamina propria[10]. ILC3s, together with other ILCs, are derived from the earliest progenitor cells (αLPs, CXCR$^+$ integrin α4β7-expressing common lymphoid progenitors (CLPs))[16], which differentiate into restricted common helper-like innate lymphoid progenitor (CHILP) cells[17]. Subsequently, downstream precursor ILCPs (common precursor of ILCs) are characterized by expression of transcription factor PLZF and can generate ILC1, ILC2, and ILC3 subsets[18]. RORγt (encoded by Rorc) drives differentiation of ILC3s from their precursor ILCPs[19,20]. RORγt deletion causes a complete loss of ILC3s but not ILC1s or ILC2s. In addition, aryl hydrocarbon receptor (AHR) is highly expressed in ILC3s, and is required for their development and maintenance[21–23]. However, the underlying mechanism that control ILC3 development and maintenance are unclear.

WASH (Wiskott-Aldrich syndrome protein (WASP) and SCAR homologue) was originally identified as an actin-nucleating factor belonging to the WASP family[24,25], and has an essential role in endosome sorting via promoting tubule fission by Arp2/3 activation. WASH is also located in autophagosomes that modulate autophagy induction[24,26]. We previously showed that WASH deficiency results in early embryonic lethality at embryonic day 7.5 (ref. 27), and its deletion in the haematopoietic system causes defective blood production of mice[28]. WASH is located in the nucleus of haematopoietic stem cells (HSCs) and promotes differentiation of haematopoietic stem cells through initiating c-Myc expression. WASH deficient T cells have been reported to have defective proliferation and impaired effector functions[29]. One study showed that WASH has a critical function in MHCII recycling and efficient priming of T helper cells[30]. However, whether WASH is involved in the development and maintenance of ILCs is unknown. Here we show that WASH is highly expressed in the nucleus of ILC3s, and associates with Arid1a to activate AHR expression. WASH-mediated AHR expression is required for the maintenance of NKp46$^+$ ILC3s, and has a critical function in their effector functions.

## Results

### WASH deficiency impairs the maintenance of NKp46$^+$ ILC3s.

We previously demonstrated that WASH is highly expressed in the nucleus of long-term hematopoietic stem cells (LT-HSCs)[28]. Conditional WASH deletion in the hematopoietic system causes defective blood production of the host 8 weeks after WASH deletion, leading to severe cytopenia and rapid anemia. To further explore whether WASH participates in the development and

maintenance of ILCs in the intestine, we assayed compositions of ILCs in Wash$^{flox/flox}$Mx1-Cre mice (gene Washc1 is referred to here as Wash) 3 weeks post poly(I:C) administration, meanwhile, bone marrow (BM) hematopoiesis was not affected by WASH deletion. We noticed that WASH deletion reduced the number of ILC3s in the intestine, but not ILC1s or ILC2s (Fig. 1a,b; Supplementary Fig. 1a,b). Of note, WASH deletion caused a greater loss of NKp46 expressing ILC3s (NKp46$^+$ ILC3s), but no impact on CD4$^+$ ILC3s or CD4$^-$NKp46$^-$ ILC3s (DN ILC3s) (Fig. 1b; Supplementary Fig. 1c). We also tested progenitors of ILC3s in the BM of Wash$^{flox/flox}$Mx1-Cre mice. We observed that WASH deletion did not affect cell numbers of their progenitors such as αLPs, CHILPs and ILCPs (Fig. 1c; Supplementary Fig. 1d–g). These observations suggest that WASH deficiency reduces the number of NKp46$^+$ ILC3s rather than other ILC cell subpopulations.

We next examined mRNA expression levels of WASH in ILC subsets. We found that WASH was actually expressed in all ILC populations and displayed a highest level in NKp46$^+$ ILC3s (Supplementary Fig. 1h,i). We then crossed Wash$^{flox/flox}$ mice with Ncr1-Cre mice[31] to generate mice with conditional deletion of WASH in NKp46$^+$ ILC3s (Fig. 1d). We observed that WASH deficiency did not affect the number of cryptopatches (Fig. 1e), suggesting that WASH is not necessary for the development of LTi cells. Since ILC3s are enriched in Peyer's patches and intestinal lamina propria[10], we then isolated lamina propria lymphocyte (LPL) populations from small and large intestines to test constitutions of ILCs. We noticed that NKp46$^+$ ILC3s were remarkably declined in the small intestine of Wash$^{flox/flox}$Ncr1-Cre mice compared with that of Wash$^{flox/flox}$ wild type (WT) control mice (Fig. 1f; Supplementary Fig. 1j). However, other ILC3 subsets, including CD4$^-$NKp46$^-$ ILC3s (DN ILC3s) and CD4$^+$ ILC3s were unchangeable in Wash$^{flox/flox}$Ncr1-Cre mice (Fig. 1f; Supplementary Fig. 1j). Similar results were achieved in both large intestines and Peyer's patches of Wash$^{flox/flox}$Ncr1-Cre mice. Given that Ncr1-Cre also induced WASH deletion in NKp46 expressing ILC1s, we then examined cell numbers of ILC1s in intraepithelial lymphocyte (IEL) populations from small intestines. Notably, WASH deletion in ILC1s did not impact cell numbers of this subset, suggesting that WASH is not required for the development and maintenance of ILC1s.

RORγt (encoded by Rorc) is involved in the development of all ILC3 subsets[11,32]. To further validate the specific role of WASH in the maintenance of NKp46$^+$ ILC3s, we generated Wash$^{flox/flox}$Rorc-Cre mice by crossing Wash$^{flox/flox}$ mice with Rorc-Cre mice to conditionally delete WASH in ILC3s (Fig. 1g). We observed that WASH deletion in ILC3s did not affect numbers of cryptopatches (Fig. 1h), confirming that WASH deficiency does not impact the development and maintenance of LTi cells. As expected, Wash$^{flox/flox}$Rorc-Cre mice significantly reduced cell numbers of NKp46$^+$ ILC3s, but not DN ILC3s or CD4$^+$ ILC3s (Fig. 1i; Supplementary Fig. 1k). Moreover, WASH deficiency had no significant effect on number changes of RORγt$^+$ T cells (Supplementary Fig. 1l). Taken together, we conclude that WASH deficiency impairs the maintenance of NKp46$^+$ ILC3s, but not their development.

Of note, Wash$^{flox/flox}$Ncr1-Cre mice did not impact the apoptosis of all ILC3 subsets (Supplementary Fig. 1m). To determine whether WASH affected the expansion of ILC3s, we then analysed cell cycle statuses of ILC3s. We found that Wash$^{-/-}$ NKp46$^+$ ILC3s maintained in a resting state compared with those of Wash$^{+/+}$ NKp46$^+$ ILC3s (Fig. 1j). Similar results were obtained from WASH deficient NKp46$^+$ ILC3s in Wash$^{flox/flox}$Rorc-Cre mice. Of note, WASH deficient mice did not cause susceptible change to Citrobacter rodentium infection (Supplementary Fig. 1n). Finally, WASH deficiency did

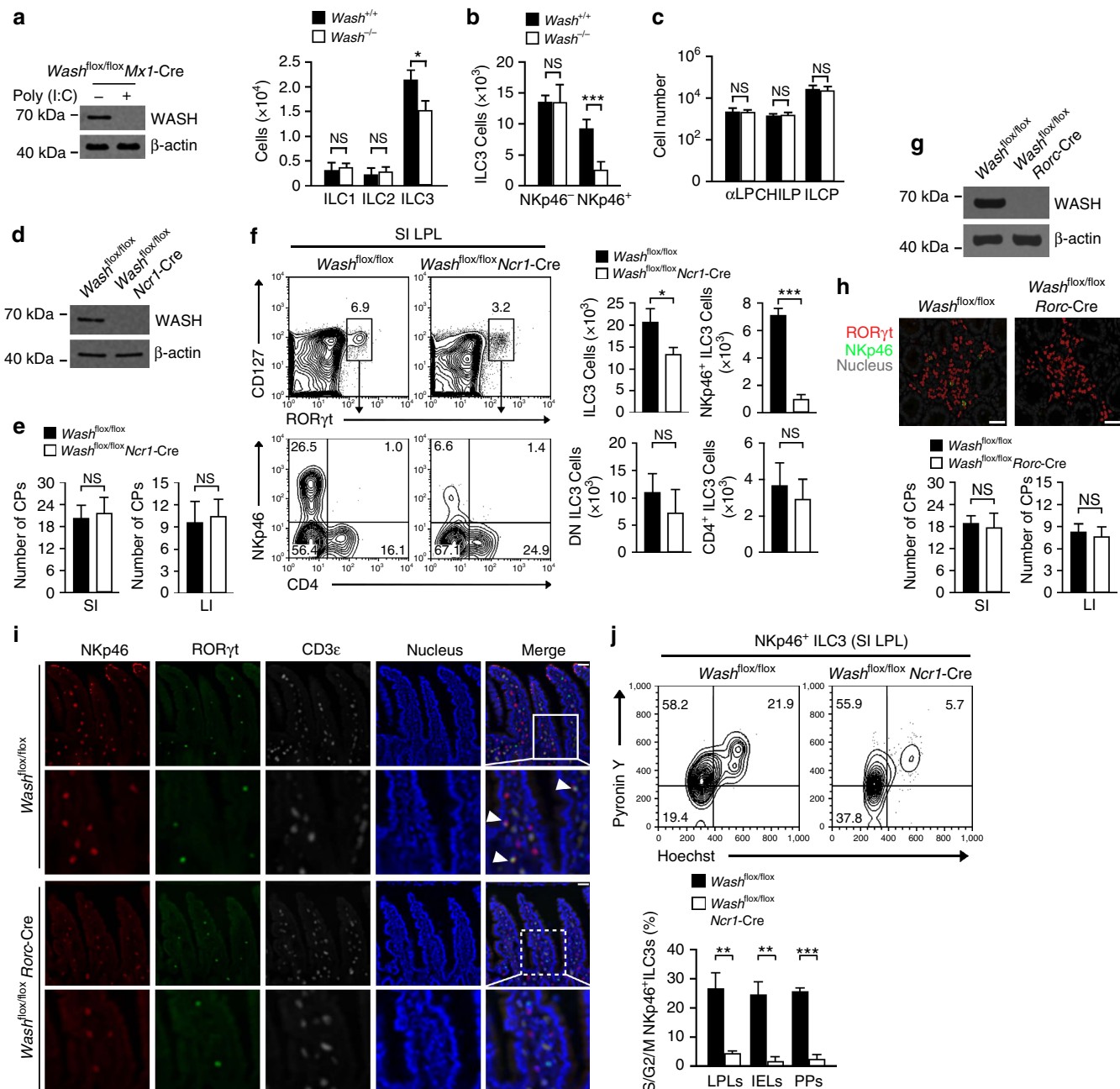

**Figure 1 | WASH deficiency reduces the cell pool of NKp46$^+$ ILC3s.** (**a**) ILCs from $Wash^{flox/flox}$ and $Wash^{flox/flox}Mx1$-Cre mice 3 weeks post poly(I:C) administration were immunoblotted with the indicated antibodies (left panel). Numbers of ILCs in small intestines of $Wash^{flox/flox}$ ($Wash^{+/+}$) and $Wash^{flox/flox}Mx1$-Cre ($Wash^{-/-}$) mice were calculated (right panel). (**b**) Numbers of NKp46$^-$ and NKp46$^+$ ILC3s in small intestines of the indicated mice were calculated. (**c**) Numbers of ILC progenitor cells in BM of the indicated mice. Gating stratedies: Lin$^-$ cKit$^{low}$ CD127$^+$α4β7$^+$ for αLP, Lin$^-$CD127$^+$α4β7$^+$ PLZF$^+$ for ILCP, Lin$^-$ CD127$^+$α4β7$^+$ CD25$^-$CD244$^+$Id2$^+$ for CHILP. For **a–c**, $n = 4$. (**d**) NKp46$^+$ ILC3s sorted from small intestine lamina propria lymphocyte populations (SI LPL) of $Wash^{flox/flox}$ and $Wash^{flox/flox}Ncr1$-Cre mice were immunoblotted with the indicated antibodies. (**e**) Numbers of cryptopatch (CP) clusters in the small intestine (left panel) or large intestine (right panel) of $Wash^{flox/flox}$ and $Wash^{flox/flox}Ncr1$-Cre mice. $n = 6$. (**f**) Flow cytometry analysis of SI LPL of $Wash^{flox/flox}$ and $Wash^{flox/flox}Ncr1$-Cre mice. Cells were gated from CD45$^+$CD19$^-$CD3$^-$ cells (left panel). Numbers of ILC3s and the indicated subpopulations of ILC3s were calculated (right panel). (**g**) ILC3s sorted from SI LPL of $Wash^{flox/flox}$ and $Wash^{flox/flox}Rorc$-Cre mice were immunoblotted with the indicated antibodies. (**h**) Representative images of SI CPs from $Wash^{flox/flox}$ and $Wash^{flox/flox}Rorc$-Cre mice staining with RORγt, NKp46 and DAPI (upper panel). Numbers of cryptopatch (CP) clusters in the small intestine (lower left) or large intestine (lower right) of $Wash^{flox/flox}$ and $Wash^{flox/flox}Rorc$-Cre mice. $n = 6$. Scale bar, 50 μm. (**i**) Small intestine sections from $Wash^{flox/flox}$ and $Wash^{flox/flox}Rorc$-Cre mice were stained with antibodies against NKp46, RORγt and CD3ε, followed by nuclear staining with DAPI. Cells with NKp46 and RORγt staining were annotated with white arrow heads. Scale bar, 100 μm. (**j**) Live NKp46$^+$ ILC3 cells were sorted from SI LPL of $Wash^{flox/flox}$ and $Wash^{flox/flox}Ncr1$-Cre mice carrying RORγt-GFP reporter through identifying surface markers, followed by Hoechst33342 and pyronin Y staining (upper panel). Percentages of S/G2/M cells in the indicated cells were calculated (lower panel). $n = 7$. Data are shown as means ± s.d. *$P < 0.05$; **$P < 0.01$; ***$P < 0.001$. Data are representative of at least three independent experiments.

not affect cell numbers of liver NK cells or NKp46$^+$ROR$\gamma$t$^-$ cells in the intestine (Supplementary Fig. 2a,b). Altogether, WASH maintains the cell pool of NKp46$^+$ ILC3 population via the regulation of cell expansion.

**WASH intrinsically maintains NKp46$^+$ ILC3s.** To examine whether WASH intrinsically affected the maintenance of NKp46$^+$ ILC3s, we transplanted $Wash^{flox/flox}Ncr1$-Cre BM cells with an equal number of CD45.1 WT BM cells into lethally irradiated CD45.1 recipient mice for 8 weeks, followed by examination of donor chimerism of ILCs post BM reconstitution (Supplementary Fig. 2c). ILC3s were significantly reduced in $Wash^{flox/flox}Ncr1$-Cre BM cells transferred mice compared with those transferred with $Wash^{flox/flox}$ control cells. By contrast, ILC1s and ILC2s did not change in $Wash^{flox/flox}Ncr1$-Cre BM cells transferred mice compared with those transferred with $Wash^{flox/flox}$ control cells. Additionally, we found that NKp46$^+$ ILC3s derived from $Wash^{flox/flox}Ncr1$-Cre donor cells were dramatically decreased in small intestines (Fig. 2a), large intestines (Fig. 2b) and Peyer's patches compared with those derived from $Wash^{flox/flox}$ donor cells. However, DN ILC3s or CD4$^+$ ILC3s derived from $Wash^{flox/flox}Ncr1$-Cre donor cells were unchangeable (Fig. 2a,b). In addition, NKp46$^+$ ILC3s derived from $Wash^{flox/flox}Ncr1$-Cre donor cells sustained in a resting state compared with those from $Wash^{flox/flox}$ donor cells (Fig. 2c; Supplementary Fig. 2d). Similar results were achieved by competitively transferring $Wash^{flox/flox}Rorc$-Cre BM cells or $Wash^{flox/flox}$ BM cells into lethally irradiated CD45.1 recipient mice (Supplementary Fig. 2e–g; Fig. 2d,e). These observations indicate that WASH deletion intrinsically affects the maintenance of NKp46$^+$ ILC3 cells.

**WASH promotes AHR expression in NKp46$^+$ ILC3s.** To explore the molecular mechanism by which WASH regulated the maintenance of NKp46$^+$ ILC3s, we screened key transcription factors and surface markers involved in development and maintenance of ILCs. We noticed that $Ahr$ was dramatically reduced in WASH deleted NKp46$^+$ ILC3s (Fig. 3a). We isolated ILC subsets from small intestines of $Wash^{flox/flox}Rorc$-Cre and $Wash^{flox/flox}$ mice and examined AHR expression. Indeed, $Ahr$ was highly expressed in NKp46$^+$ ILC3s derived from $Wash^{flox/flox}$ mice (Fig. 3b), with a modest expression in DN ILC3s and CD4$^+$ ILC3s. These results were further validated by immunoblotting (Supplementary Fig. 2h). Notably, WASH knockout specifically impeded AHR expression in NKp46$^+$ ILC3s, but not in DN ILC3s or CD4$^+$ ILC3s (Fig. 3b), suggesting WASH is involved in the regulation of AHR expression in NKp46$^+$ ILC3s. Through chromatin immunoprecipitation (ChIP) assays, we found that WASH bound to $Ahr$ promoter ($-400$ to $-200$) in NKp46$^+$ ILC3s (Fig. 3c), but not in DN ILC3s or CD4$^+$ ILC3s (Fig. 3d). WASH deficiency markedly suppressed $Ahr$ transcription in NKp46$^+$ ILC3s by a nuclear run-on assay, but not in DN ILC3s or CD4$^+$ ILC3s (Fig. 3e). We then transplanted WASH over-expressing BM cells together with recipient BM cells into lethally irradiated CD45.1 recipient mice for reconstitution assays. We observed that WASH overexpression augmented $Ahr$ transcription in NKp46$^+$ ILC3s, but not in DN ILC3s or CD4$^+$ ILC3s (Fig. 3f), suggesting other factors than WASH may be required for $Ahr$ expression in DN ILC3s or CD4$^+$ ILC3s. These data indicate that WASH promotes AHR expression in NKp46$^+$ ILC3s through association with its promoter.

We previously showed that WASH acts as a transcription associating factor to promote transcription of target genes via its

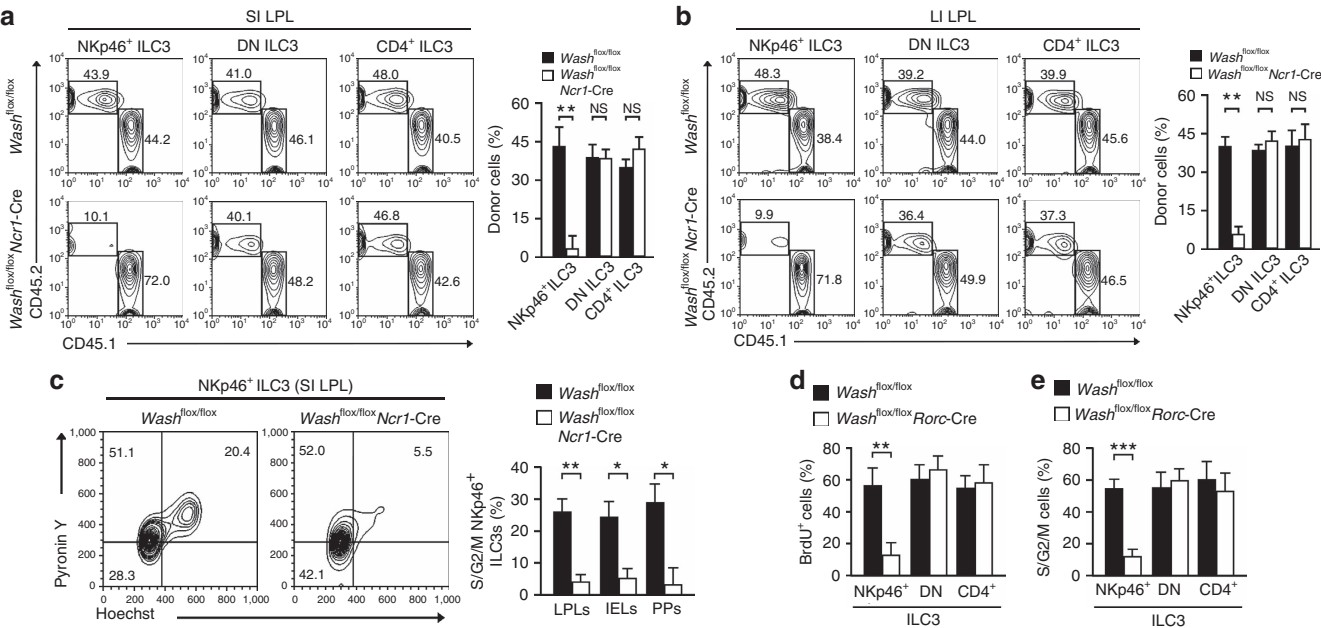

**Figure 2 | WASH affects the maintenance of NKp46$^+$ ILC3s in a cell-intrinsic manner.** (**a,b**) $2 \times 10^6$ $Wash^{flox/flox}$ROR$\gamma$t-GFP or $Wash^{flox/flox}Ncr1$-CreROR$\gamma$t-GFP BM cells were co-transplanted with $2 \times 10^6$ CD45.1 BM cells into lethally irradiated CD45.1 recipient mice, followed by donor chimerism examination of the indicated ILC3 subpopulations in small intestine (**a**) and large intestine (**b**) 8 wk later through flow cytometry (left panel). Donor chimerisms were calculated (right panel). For **a** and **b**, $n = 6$. (**c**) Live NKp46$^+$ ILC3 cells were sorted from mice reconstituted as in **a** through identifying surface markers, followed by Hoechst33342 and pyronin Y staining (left panel). Percentages of S/G2/M cells in the indicated cells were calculated (right panel). $n = 5$. (**d**) Mice reconstituted with $Wash^{flox/flox}$ROR$\gamma$t-GFP or $Wash^{flox/flox}Rorc$-CreROR$\gamma$t-GFP BM cells were intraperitoneally injected with 7 mg kg$^{-1}$ BrdU for 16 h, followed by flow cytometry analysis of BrdU signals in the indicated ILC3 cells. Percentages of BrdU positive cells were calculated. $n = 4$. (**e**) Live ILC3 cells were sorted from mice reconstituted with $Wash^{flox/flox}$ROR$\gamma$t-GFP or $Wash^{flox/flox}Rorc$-CreROR$\gamma$t-GFP BM cells through identifying surface markers, followed by Hoechst33342 and pyronin Y staining. Percentages of S/G2/M cells in the indicated cells were calculated. $n = 4$. Data are shown as means ± SD. *$P < 0.05$; **$P < 0.01$; ***$P < 0.001$. Data are representative of at least three independent experiments.

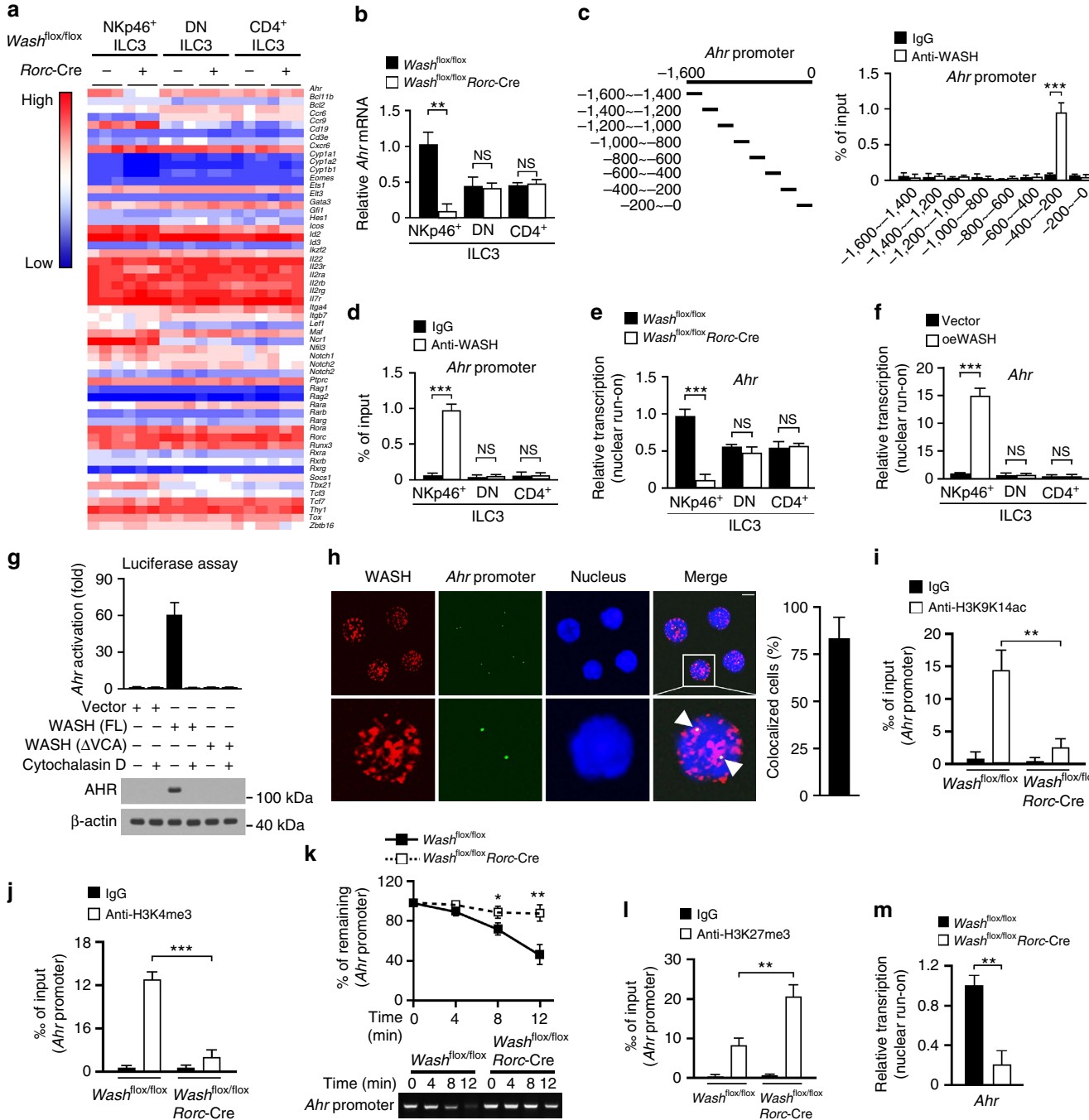

**Figure 3 | WASH promotes AHR expression in NKp46$^+$ ILC3s.** (**a**) Expression levels of the indicated genes were examined in the indicated cells sorted from the indicated mice by RT-PCR analysis. (**b**) AHR expression levels were examined in the indicated ILC3 subsets from the indicated mice. For **a** and **b**, n = 5. (**c**) Sorted NKp46$^+$ ILC3 cells were subjected to ChIP assay with antibody against WASH, followed by detection of *Ahr* promoter (right panel) with different primers shown as in the left panel. (**d**) The indicated ILC3 subsets were subjected to ChIP assay with anti-WASH antibody, followed by detection of *Ahr* promoter through PCR. (**e**) The indicated ILC3 subsets were subjected to nuclear run-on assay, followed by RT-PCR analysis of *Ahr*. (**f**) Vector or WASH overexpressing ILC3 subsets were subjected to nuclear run-on assay, followed by RT-PCR analysis of *Ahr*. For **c**–**f**, n = 5. (**g**) pGL3-AHR promoter expressing NK92 cells were transfected with the indicated WASH variants with or without cytochalasin D, followed by luciferase assays (upper panel). Endogenous expression levels of AHR were examined by immunoblotting with anti-AHR antibody (lower panel). (**h**) *In situ* hybridization of *Ahr* promoter and WASH in sorted NKp46$^+$ ILC3 cells (left panel). White arrow head denotes *Ahr* promoters colocalized with WASH. Percentages of cells with WASH colocalized with *Ahr* promoter were calculated (right panel). At least 200 NKp46$^+$ ILC3 cells were counted. Scale bar, 5 μm. (**i**,**j**) ChIP analysis of *Ahr* promoter in NKp46$^+$ ILC3 cells sorted from the indicated mice with antibodies against H3K9K14ac (**i**) or H3K4me3 (**j**). (**k**) NKp46$^+$ ILC3 nuclei of the indicated mice were treated with indicated units of DNase I. DNA was extracted and examined by PCR (lower panel). Intensities of PCR products were calculated (upper panel). (**l**) ChIP analysis of *Ahr* promoter in NKp46$^+$ ILC3 cells sorted from the indicated mice with anti-H3K27me3 antibody. (**m**) NKp46$^+$ ILC3 cells sorted from the indicated mice were subjected to nuclear run-on assay, followed by RT-PCR analysis of *Ahr*. For (**i**–**m**), n = 9. Data are shown as means ± s.d. *P < 0.05; **P < 0.01; ***P < 0.001. Data are representative of at least three independent experiments.

VCA domain[28]. We next wanted to determine whether the VCA domain of WASH was required for *Ahr* transcription. We then transfected full-length WASH (WASH(FL)) or VCA truncated WASH (WASH(ΔVCA)) into pGL3-AHR expressing NK92 cells for luciferase assays. Consistent with our previous results, WASH(ΔVCA) abrogated *Ahr* activation, whereas WASH(FL) was able to activate *Ahr* transcription (Fig. 3g). The actin nucleation inhibitor cytochalasin D blocked *Ahr* activation (Fig. 3g). Furthermore, WASH was co-localized with *Ahr* promoter in NKp46[+] ILC3s by fluorescence staining (Fig. 3h). Furthermore, WASH deficiency repressed the acetylation of H3K9K14 and the methylation of H3K4 on *Ahr* promoter (Fig. 3i,j), both of which are hallmarks of active gene transcription. Additionally, WASH knockout also made *Ahr* promoter more resistant to DNase I digestion (Fig. 3k). Consistently, the *Ahr* promoter region accumulated more repressive histone markers in WASH deficient NKp46[+] ILC3s (Fig. 3l). Finally, *Ahr* activation was remarkably suppressed in WASH deficient NKp46[+] ILC3s (Fig. 3m). These observations confirm that WASH promotes *Ahr* transcription.

To further validate that WASH regulated the maintenance of NKp46[+] ILC3s via AHR, we rescued AHR expression in WASH deficient cells by transplanting AHR overexpressing *Wash*[flox/flox]*Rorc*-Cre BM cells together with recipient BM cells into lethally irradiated recipient mice (Supplementary Fig. 3a). Consequently, AHR overexpression rescued cell numbers of NKp46[+] ILC3s reduced by WASH knockout (Supplementary Fig. 3b). Additionally, AHR restoration in NKp46[+] ILC3s maintained more cycling cells compared with those with empty vector expression (Supplementary Fig. 3c). We also rescued WASH(FL) or WASH(ΔVCA) in WASH deficient NKp46[+] ILC3s by transplanting WASH(FL) or WASH(ΔVCA) overexpressing *Wash*[flox/flox]*Rorc*-Cre BM cells together with recipient BM cells into lethally irradiated recipient mice (Supplementary Fig. 3d). WASH(FL) overexpression was able to restore the cell number of NKp46[+] ILC3s reduced by WASH deletion, whereas WASH(ΔVCA) has no such effect (Supplementary Fig. 3e). Consistently, WASH restoration promoted AHR expression in NKp46[+] ILC3s (Supplementary Fig. 3f), and sustained more cycling cells (Supplementary Fig. 3g). Of note, anti-Arid1a antibody could precipitate *Ahr* promoter in anti-WASH antibody precipitates (Supplementary Fig. 3h). Anti-WASH antibody could also precipitate the *Ahr* promoter in anti-Arid1a antibody precipitates (Supplementary Fig. 3i), suggesting that WASH and Arid1a together bind to the *Ahr* promoter region. Furthermore, WASH associated with Arid1a only in the NKp46[+] ILC3s (Supplementary Fig. 3j), suggesting that the *Ahr* expression is differentially regulated among different ILC3 subsets. Altogether, WASH maintains the cell pool of NKp46[+] ILC3s via promoting AHR expression.

**WASH associates with Arid1a to promote *Ahr* transcription.** To elucidate how WASH promoted *Ahr* transcription in NKp46[+] ILC3s, we screened a cDNA library using WASH as bait via a yeast two-hybrid system. We identified Arid1a as a new interactor of WASH (Fig. 4a). Arid1a belongs to the BRG1-associated factor (BAF) complex that is involved in nucleosome remodelling and gene transcription[33]. Recombinant WASH could precipitate Arid1a from LPL lysates of small intestine (Fig. 4b). Moreover, anti-WASH antibody could precipitate Arid1a from lysates of NKp46[+] ILC3s (Fig. 4c), confirming the interaction of WASH with Arid1a. By contrast, we noticed that Arid1a signals could not be picked up by anti-Arid1a antibody in anti-WASH precipitates derived from Arid1a deleted ILC3s lysates (Fig. 4c). These results validated the specificity of these two antibodies we

used. Through domain mapping, we identified that two fragments of Arid1a (aa 968–1,484 and aa 1,935–2,283) were required for WASH binding (Fig. 4d). Of note, Arid1a knockdown abolished the interaction between WASH and other components of BAF complex (Fig. 4e), suggesting that WASH associates with BAF complex via Arid1a interaction. Similar to WASH, Arid1a bound to the same region of *Ahr* promoter in NKp46[+] ILC3s (Fig. 4f). Additionally, other components of BAF complex were also recruited to *Ahr* promoter (Fig. 4g). These data suggest that WASH may recruit BAF complex to *Ahr* promoter for enhancing its transcription.

We generated *Arid1a*[flox/flox]*Rorc*-Cre mice by crossing *Arid1a*[flox/flox] mice with *Rorc*-Cre mice. Arid1a was completely deleted in ILC3s (Fig. 4h). Through nuclear run-on assays, we noticed that Arid1a knockout repressed *Ahr* transcription in NKp46[+] ILC3s (Fig. 4i), but not in DN ILC3s or CD4[+] ILC3s. We next transplanted Arid1a overexpressing BM cells together with recipient BM cells into lethally irradiated recipient mice. We observed that Arid1a overexpression caused elevated expression of AHR in NKp46[+] ILC3s, but not in DN ILC3s or CD4[+] ILC3s (Fig. 4j). We observed that Arid1a was expressed in three subsets of ILC3s (Fig. 5a). Moreover, Arid1a deficiency suppressed the acetylation of H3K9K14 and the methylation of H3K4 on *Ahr* promoter (Fig. 5b,c). Arid1a knockout also made *Ahr* promoter more resistant to DNase I cleavage (Fig. 5d), suggesting an inaccessible state of *Ahr* promoter for transcriptional activation. We next transfected full-length Arid1a (Arid1a(FL)) or two Arid1a mutants lacking WASH binding domains into pGL3-AHR expressing WASH silenced NK92 cells, followed by examination of *Ahr* activation through luciferase assays. We observed that Arid1a mutants failed to activate *Ahr* transcription in both shCtrl treated and WASH silenced cells (Fig. 5e). By contrast, Arid1a(FL) could activate *Ahr* transcription in shCtrl treated cells, but not in WASH silenced cells (Fig. 5e). These observations suggest that Arid1a promotes *Ahr* transcription in a WASH-dependent manner.

WASH knockout abrogated the co-localization of Arid1a with *Ahr* promoter in NKp46[+] ILC3s by fluorescence imaging (Fig. 5f). In parallel, WASH deletion impaired the recruitment of Arid1a and BAF complex to *Ahr* promoter by ChIP assays (Fig. 5g,h). Finally, in WASH silenced cells, Arid1a mutants failed to bind *Ahr* promoter (Fig. 5i). Taken together, we conclude that WASH recruits the BAF complex to promote AHR expression in NKp46[+] ILC3s.

**Arid1a maintains NKp46[+] ILC3s through AHR ILC3s.** We next examined ILC3 subsets in *Arid1a*[flox/flox]*Rorc*-Cre and *Arid1a*[flox/flox] mice. We observed that *Arid1a*[flox/flox]*Rorc*-Cre mice showed reduced numbers of NKp46[+] ILC3s, but not other subsets of ILC3s (Fig. 6a), these observations were similar to those of *Wash*[flox/flox]*Rorc*-Cre mice. In addition, Arid1a deficiency did not affect the apoptosis of NKp46[+] ILC3s either. We found that Arid1a deficient NKp46[+] ILC3s derived from *Arid1a*[flox/flox]*Rorc*-Cre mice incorporated much less BrdU than those from *Arid1a*[flox/flox] mice (Fig. 6b). In addition, *Arid1a*[−/−] NKp46[+] ILC3s maintained in a resting state compared with those of *Arid1a*[+/+] NKp46[+] ILC3s (Fig. 6c). Similar results were obtained when we reconstituted recipient mice with Arid1a deficient BM cells. We next overexpressed full length or mutant Arid1a in *Arid1a*[flox/flox]*Rorc*-Cre BM cells and then transplanted these BM cells together with recipient BM cells into lethally irradiated recipient mice. We found that overexpression of two Arid1a mutants lacking WASH binding ability in *Arid1a*[flox/flox]*Rorc*-Cre BM cells failed to rescue donor reconstitution ratios of NKp46[+] ILC3s caused by Arid1a deficiency (Fig. 6d). By contrast,

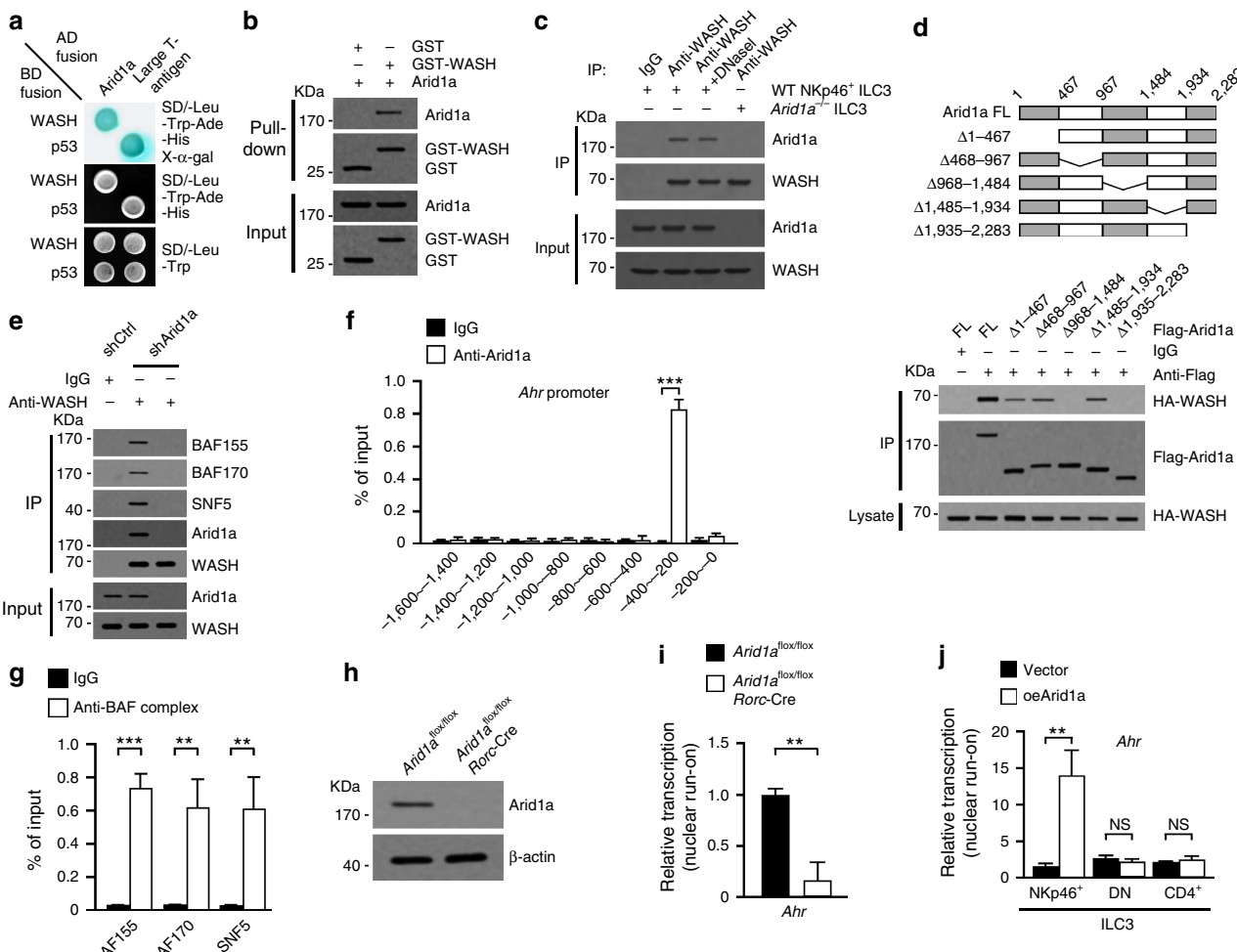

**Figure 4 | WASH associates with Arid1a.** (**a**) Yeast strain AH109 was co-transfected with Gal4 DNA-binding domain (BD)-fused WASH and Gal4 activating domain (AD)-fused Arid1a. p53 and large T antigen were introduced as a positive control. (**b**) GST-WASH was incubated with LPL lysates, followed by a GST pulldown assay. (**c**) $4 \times 10^5$ NKp46$^+$ ILC3 cells (pooled from RORγt-GFP reporter mice) were lysed and immunoprecipitated with anti-WASH antibody, followed by immunoblotting with the indicated antibodies. Lysates were treated by DNase I before incubation with antibodies. $4 \times 10^5$ Arid1a deleted ILC3 cells sorted from Arid1a$^{flox/flox}$Rorc-Cre mice were also lysed and immunoprecipitated with anti-WASH antibody for co-IP assay as a control. Experiment was repeated for two times. (**d**) Flag-tagged WT and truncated Arid1a (upper panel) were co-transfected with HA-tagged WASH into NK92 cells, followed by immunoprecipitation with antibody against Flag. Immunoprecipitates were immunoblotted with the indicated antibodies (lower panel). (**e**) NK92 cells with or without Arid1a knockdown were subjected to immunoprecipitation with antibody against WASH, followed by immunoblotting with the indicated antibodies. (**f**) Sorted NKp46$^+$ ILC3 cells were subjected to ChIP assay with antibody against Arid1a, followed by PCR analysis with the indicated fragment primers of Ahr. (**g**) Sorted NKp46$^+$ ILC3 cells were subjected to ChIP assay with antibodies against the indicated BAF components, followed by PCR examination with primers specific to Ahr promoter. (**h**) ILC3s were sorted from Arid1a$^{flox/flox}$ and Arid1a$^{flox/flox}$Rorc-Cre mice, followed by immunoblotting. (**i**) NKp46$^+$ ILC3 cells from Arid1a$^{flox/flox}$ and Arid1a$^{flox/flox}$Rorc-Cre mice were subjected to nuclear run-on assay, followed by RT-PCR analysis of Ahr. (**j**) Arid1a overexpressing BM cells together with equal numbers of recipient BM cells were transplanted into lethally irradiated recipient mice. Vector or Arid1a overexpressing ILC3 cells were sorted from above reconstituted mice and subjected to nuclear run-on assay, followed by RT-PCR analysis of Ahr. For **i** and **j**), $n = 5$. Data are shown as means ± s.d. *$P < 0.05$; **$P < 0.01$; ***$P < 0.001$. Data represent at least three independent experiments unless mentioned.

overexpression of full length Arid1a in Arid1a$^{flox/flox}$Rorc-Cre BM cells could rescue donor reconstitution rates of NKp46$^+$ ILC3s (Fig. 6d). Consistently, Arid1a restoration promoted AHR expression in NKp46$^+$ ILC3s (Fig. 6e), and sustained more cycling cells of these NKp46$^+$ ILC3s (Fig. 6f). These results indicate that Arid1a is essential for the maintenance of NKp46$^+$ ILC3s.

We next wanted to determine whether WASH and Arid1a exerted a synergistic effect. We generated WASH and Arid1a double knockout (DKO) mice through crossing Arid1a$^{flox/flox}$Rorc-Cre mice with Wash$^{flox/flox}$ mice. We noticed that DKO mice did not affect numbers of Peyer's patches. Consistently, DKO mice markably decreased cell numbers of

NKp46$^+$ ILC3s as well as IL-22 expressing ILC3s (Supplementary Fig. 4a, b). In addition, DKO mice dramatically reduced AHR expression in NKp46$^+$ ILC3s (Supplementary Fig. 4c), accompanied with fewer cycling cells (Supplementary Fig. 4d). Finally, DKO mice did not impact apoptosis of NKp46$^+$ ILC3s (Supplementary Fig. 4e). These observations indicate that WASH and Arid1a exert synergistc effect on Ahr expression.

We next rescued AHR in Arid1a KO or DKO NKp46$^+$ ILC3s by transplanting AHR overexpressing Arid1a$^{flox/flox}$Rorc-Cre or Wash$^{flox/flox}$Arid1a$^{flox/flox}$Rorc-Cre BM cells together with recipient BM cells into lethally irradiated recipient mice for reconstitution assays (Supplementary Fig. 4f). AHR restoration

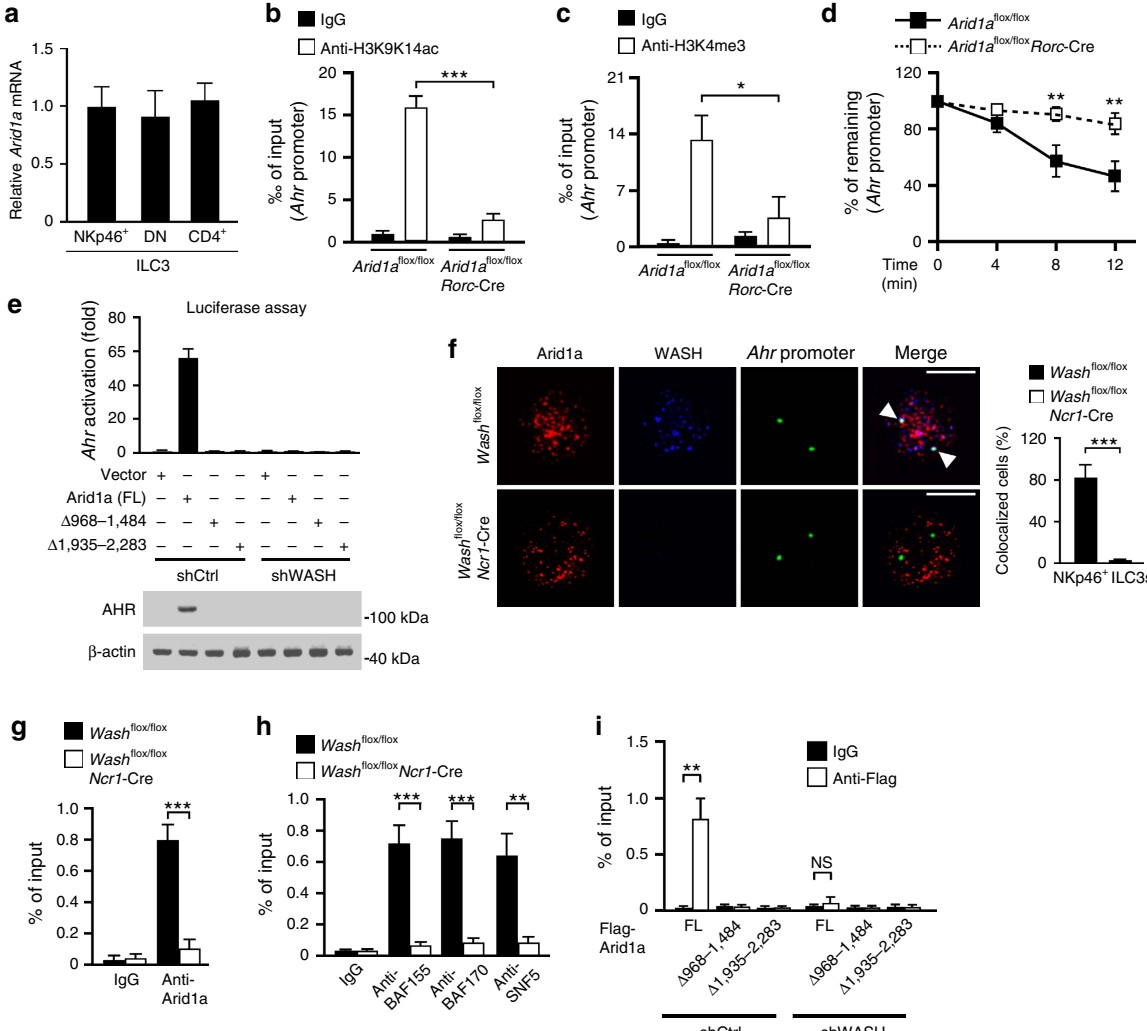

**Figure 5 | WASH is required for Arid1a to activate AHR expression.** (**a**) Arid1a mRNA levels in the indicated cell subsets were examined through RT-PCR. (**b,c**) ChIP analysis of *Ahr* promoters in NKp46$^+$ ILC3 cells from *Arid1a*$^{flox/flox}$ and *Arid1a*$^{flox/flox}$*Rorc*-Cre mice with antibodies against H3K9K14ac (**b**) or H3K4me3 (**c**). (**d**) Nuclei isolated from *Arid1a*$^{flox/flox}$ and *Arid1a*$^{flox/flox}$*Rorc*-Cre NKp46$^+$ ILC3 cells were digested by 1 unit of DNase I for the indicated times, followed by DNA extraction for PCR analysis of *Ahr* promoter. For **b**–**d**, *n* = 6. (**e**) WASH silenced NK92 cells were transfected with full-length (FL) or truncated Arid1a (Δ968-1484 and Δ1935-2283), followed by luciferase assay (upper panel). Endogenous expression levels of AHR were examined by immunoblotting with anti-AHR antibody (lower panel). (**f**) NKp46$^+$ ILC3 cells from *Wash*$^{flox/flox}$ and *Wash*$^{flox/flox}$*Ncr1*-Cre mice were *in situ* hybridized with probes against *Ahr* promoter, followed by staining with antibodies against WASH and Arid1a (upper panel). White arrow head indicates *Ahr* promoters colocalized with Arid1a. Percentages of cells with Arid1a colocalized *Ahr* promoter were calculated (lower panel). At least 200 NKp46$^+$ ILC3 cells were counted. Scale bar, 5 μm. (**g**) NKp46$^+$ ILC3 cells from *Wash*$^{flox/flox}$ and *Wash*$^{flox/flox}$*Ncr1*-Cre mice were subjected to ChIP assay with antibody against Arid1a, followed by PCR assay of *Ahr* promoter. *n* = 5. (**h**) NKp46$^+$ ILC3 cells from *Wash*$^{flox/flox}$ and *Wash*$^{flox/flox}$*Ncr1*-Cre mice were subjected to ChIP assay with antibodies against the indicated BAF components, followed by PCR assay of *Ahr* promoter. *n* = 5. (**i**) Control and WASH silenced NK92 cells were transfected with the indicated Arid1a plasmids, followed by ChIP of *Ahr* promoter with antibody against Flag. Data are shown as means ± s.d. *P<0.05; **P<0.01; ***P<0.001. Data are representative of at least three independent experiments.

rescued donor reconstitution ratios of NKp46$^+$ ILC3s reduced by Arid1a KO or DKO of WASH with Arid1a (Supplementary Fig. 4g), and maintained more cycling cells (Supplementary Fig. 4h). We next transplanted Arid1a-rescued *Arid1a*$^{flox/flox}$-*Rorc*-Cre or *Wash*$^{flox/flox}$*Arid1a*$^{flox/flox}$*Rorc*-Cre BM cells together with recipient BM cells into lethally irradiated recipient mice. Consistently, Arid1a overexpression could indeed rescue donor reconstitution ratios of NKp46$^+$ ILC3s reduced by Arid1a KO or DKO (Supplementary Fig. 5a,b), and sustained more cycling cells (Supplementary Fig. 5c). Consequently, Arid1a overexpression augmented AHR expression in *Arid1a*$^{-/-}$ NKp46$^+$ ILC3s, but not in *Wash*$^{-/-}$*Arid1a*$^{-/-}$ NKp46$^+$ ILC3s (Supplementary Fig. 5d). In addition, AHR ligand FICZ engagement sustained more cycling NKp46$^+$ ILC3s in WT ILC3s, but not in *Arid1a*$^{-/-}$ or in *Wash*$^{-/-}$*Arid1a*$^{-/-}$ ILC3s (Supplementary Fig. 5e). By contrast, AHR antagonist treatment rendered NKp46$^+$ ILC3s in resting state even in WT ILC3s (Supplementary Fig. 5e). Altogether, WASH-mediated AHR expression is required for the maintenance of NKp46$^+$ ILC3s.

## Discussion

Innate lymphoid cells (ILCs) are a distinct arm of the innate immune system, and can directly communicate with other hematopoietic and non-hematopoietic cells to regulate immunity, inflammation and tissue homeostasis[34]. However, how these ILC

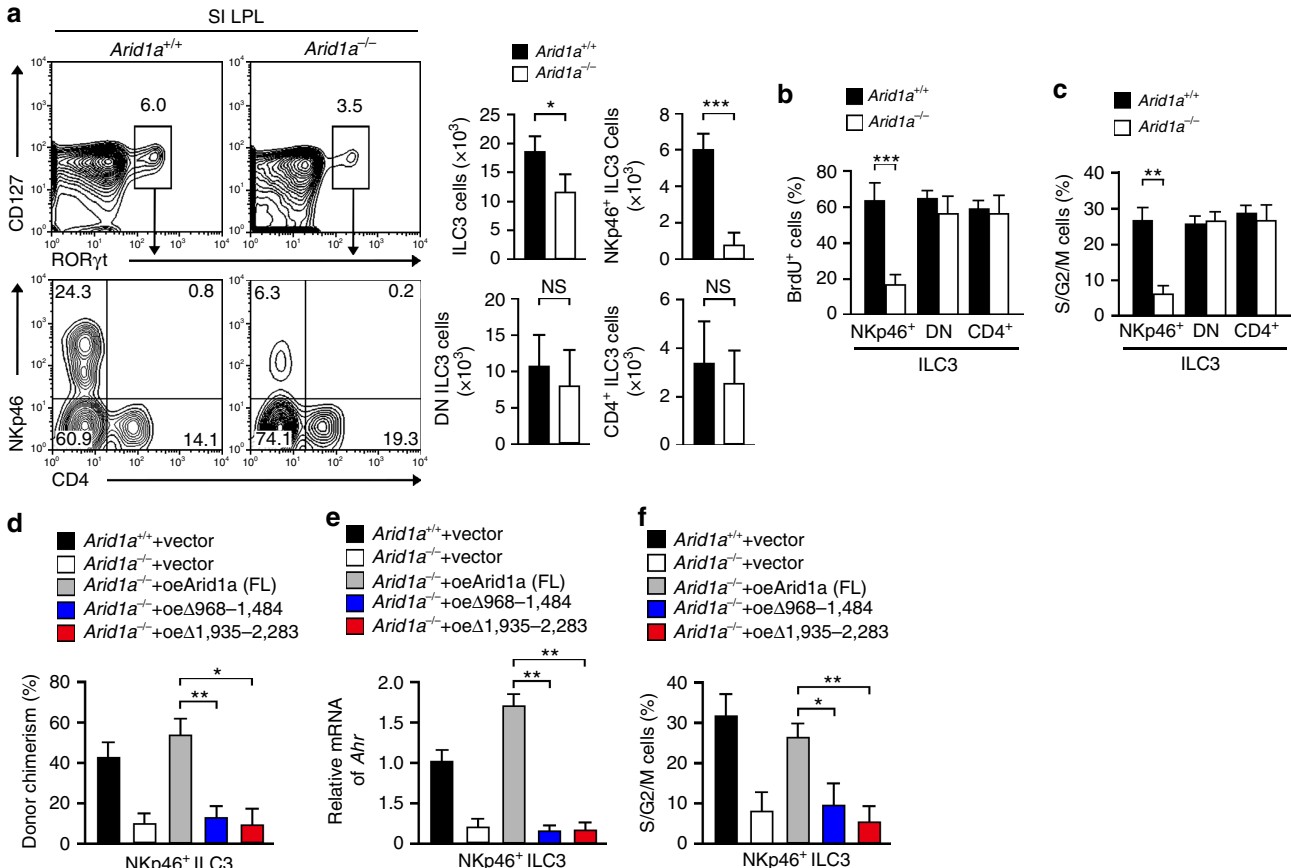

**Figure 6 | Arid1a deficiency impedes AHR expression.** (**a**) Flow cytometry analysis of SI LPLs of Arid1a^flox/flox and Arid1a^flox/flox Rorc-Cre mice (referred to as Arid1a^+/+ and Arid1a^−/−). Cells were gated from CD45^+CD19^−CD3^− cells (left panel). Numbers of ILC3s and the indicated subpopulations of ILC3s were calculated (right panel). (**b**) Arid1a^flox/flox and Arid1a^flox/flox Rorc-Cre mice were intraperitoneally injected with 7 mg kg^−1 BrdU for 16 h, followed by flow cytometry analysis of BrdU signals in the indicated ILC3 cells. Percentages of BrdU positive cells were calculated. $n = 4$. (**c**) Live ILC3 cells were sorted from Arid1a^flox/flox;RORγt-GFP and Arid1a^flox/flox Rorc-Cre;RORγt-GFP mice through identifying surface markers, followed by Hoechst33342 and pyronin Y staining. Percentages of S/G2/M cells in the indicated cells were calculated. $n = 3$. (**d–f**) Arid1a^flox/flox and Arid1a^flox/flox Rorc-Cre BM cells transfected with full-length Arid1a or mutant Arid1a were co-transplanted with equal numbers of CD45.1 recipient BM cells into lethally irradiated mice, followed by examination of donor chimerism (**d**), AHR expression (**e**) and cell cycle (**f**) of NKp46^+ ILC3s 8 week later. For **d–f**, $n = 7$. Data are shown as means ± s.d. *$P < 0.05$; **$P < 0.01$; ***$P < 0.001$. Data are representative of at least three independent experiments.

lineages develop and maintain remains largely unknown. In this study, we show that WASH is highly expressed in the nucleus of ILC3s. WASH deletion impairs the maintenance of NKp46^+ ILC3s but not other ILC subsets. In NKp46^+ ILC3s, WASH recruits Arid1a to Ahr promoter to activate AHR expression. WASH-mediated AHR expression is required for the maintenance of NKp46^+ ILC3s. WASH deletion in ILC3s reduced the number of NKp46^+ ILC3s. Notably, double knockout of WASH and Arid1a impedes AHR expression and NKp46^+ ILC3s maintenance.

All ILC subsets are derived from common lymphoid progenitors (CLPs), which also differentiate into T and B cells[17,34]. The earliest progenitor cells specific to ILCs are CXCR^+ integrin α4β7–expressing CLPs, referred to as α–lymphoid precursor (αLP) cells, which give rise to ILC1, ILC2, ILC3 and conventional NK cells (cNKs)[35,36]. The common progenitor to all ILC lineages (CHILP) is identified as its Lin^−CD127^+Id2^+CD25^−α4β7^+ phenotype and differentiates to all ILC subsets, but not cNKs[17]. The common precursor to ILCs (ILCP) is defined by expression of transcription factor PLZF and gives rise to ILC1, ILC2, and ILC3 subpopulations[18]. We previously reported that conditional deletion of WASH in the hematopoietic system impairs the differentiation of HSCs and

causes defective blood production of the host[28]. In this study, we show that WASH deletion in the hematopoietic system does not impact the development of progenitors of ILCs, including αLP, CHILP and ILCP cells. However, WASH deficiency only affects the maintenance of NKp46^+ ILC3 subsets, but not other ILC lineages or ILC3 subpopulations. Our observations suggest that the development and maintenance of different ILC subsets utilize different intrinsic checkpoints.

Group 3 ILCs (ILC3) are defined by their expression of RORγt, including CCR6^+ lymphoid tissue inducer (LTi) cells and CCR6^− ILC3 lineage[15,37]. Through expression of lymphotoxin (LT) α1β2, LTi cells induce the development of lymph nodes and Peyer's patches through LTβ receptor (LTβR) signalling[38,39]. Some CCR6^− ILC3s express a natural cytotoxicity-triggering receptor (NCR): NKp46 in mice[32] and NKp44 in humans[40], and eventually develop into NKp46^+ ILC3s. Unlike LTi cells, CCR6^− ILC3s are developed from PLZF^+ ILCP cells[18], and they proliferate largely in mice mainly after birth to 3–4 weeks of age[41]. NKp46^+ ILC3s uniquely produces only IL-22, not IL-17 (refs 14,15), and have the potential to differentiate into IFN-γ producing ILC1s[17]. Here we show that WASH deletion only impairs the maintenance of NKp46^+ ILC3s and reduces IL-22 production. However, WASH deletion did not change the IFN-γ

secretion, suggesting that WASH specifically controls the maintenance of NKp46$^+$ ILC3s, but not IFN-γ producing ILC1s.

WASH, a member of WASP family, plays a critical role in endosomal trafficking via its actin nucleation function[24,25,42]. We previously showed that WASH is localized at autophagosomes of mouse embryonic fibroblasts (MEFs) and regulates autophagy induction[26,27]. We also found that WASH is constitutively expressed in the blood cells and mainly resides in the nucleus of long-term hematopoietic stem cells (LT-HSCs)[28]. WASH deletion breaks the balance of self-renewal and differentiation of HSCs. We demonstrated that WASH promotes the differentiation of LT-HSCs through association with the NURF complex, which activates c-Myc transcription[28]. Chromatin remodelling complexes are involved in nucleosome sliding, dissociation, or replacement using the energy derived from ATP hydrolysis[43,44]. Herein we show that WASH is located in the nucleus of NKp46$^+$ ILC3s and associates with the BAF chromatin remodelling complex through direct interaction with Arid1a. Arid1a, also known as BAF250a, is a major component of BAF complex[45]. Arid1a deficiency impairs the pluripotency and self-renewal of embryonic stem (EC) cells, and results in early embryonic lethality[46]. Here we showed that WASH is preferentially highly expressed in NKp46$^+$ ILC3s. High nuclear levels of WASH in NKp46$^+$ ILC3s could be able to recruit the BAF complex to Ahr promoter for promoting its transcription. WASH-mediated AHR expression controls the maintenance of NKp46$^+$ ILC3 cell pool of the host. However, why WASH is preferentially expressed in NKp46$^+$ ILC3s needs to be further investigated.

AHR is expressed in many cell types and thus exerts pleiotropic effects by integrating with other signalling pathways. AHR deficient mice frequently died after birth or exhibited a slower growth rate in the first few weeks of life[47]. AHR signalling is implicated in a variety of immune responses from different cell types. Especially, AHR plays a critical role at mucosal interfaces where the host encounters other living entities and environmental agents. Of note, ILCs are preferentially distributed at mucosal surfaces, where they can perceive changes in the surrounding microenvironment primarily via cytokine receptor signalling[14,23]. ILC3s express high levels of AHR and sensitively respond to AHR signalling. AHR signalling is not only needed for IL-22 production by ILC3s, but is necessary for their development and/or maintenance[21–23]. AHR deficient mice displayed reduced numbers of ILC3s in small intestines and in Peyer's patches. Here we show that WASH-mediated AHR expression controls the maintenance of NKp46$^+$ ILC3s, but not other ILC lineages. Additionally, WASH-mediated AHR expression has no significant impact on the development of NKp46$^+$ ILC3s. Consequently, decreased numbers of NKp46$^+$ ILC3s substantially reduce production of IL-22. In sum, WASH-mediated AHR expression is indispensable for the maintenance of NKp46$^+$ ILC3s in the intestine.

## Methods

**Antibodies and reagents.** A rabbit polyclonal antibody against WASH was generated from the VCA domain of WASH protein as described previously[27]. Commercial antibodies: Antibodies against CD3 (17A2), CD4 (GK1.5), CD19 (1D3), CD127 (A7R34), c-Kit (2B8), Sca-1 (D7), CD25 (PC61.5), CD11b (M1/70), CD11c (N418), Gr1 (RB6-8C5), F4/80 (BM8), Ter119 (TER-119), CD27 (LG.7F9), CD90 (HIS51), CD45.2 (104), RORγt (AFKJS-9), NKp46 (29A1.4), CD244 (C9.1), Flt3 (A2F10), Integrinα4β7 (DATK32), CD45.1 (A20), NK1.1 (PK136), IL-22 (IL22JOP), Thy1.2 (30-H12), and PLZF (Mags.21F7) were purchased from eBiosciences (San Diego, USA) and used in a 1:100 dilution for flow cytometric staining. Anti-Arid1a (D2A8U), anti-BAF155 (D7F8S), anti-BAF170 (D8O9V) and anti-SNF5 (D8M1X) antibodies were from Cell Signaling Technology and used in a 1:2,000 dilution for werstern blotting and in a 1:500 dilution for immunofluorescence staining. Anti-GST (6G9C6) and anti-β-actin (SP124) antibodies were from Sigma-Aldrich and used in a 1:2,000 dilution for western blotting. Donkey anti-rabbit IgG secondary antibodies conjugated with Alexa-488, Alexa-594 or Alexa-405 were purchased from Molecular Probes. Donkey anti-mouse IgG secondary antibodies conjugated with Alexa-488 or Alexa-594 were from Molecular Probes. Anti-HA and HRP-conjugated secondary antibodies were from Santa Cruz. BrdU, DAPI and Cytochalasin D were from Sigma-Aldrich. FICZ was from Enzo Life Sciences. AHR antagonist was from Millipore. IL-22 detection kit was from eBioscience.

**Cell culture and transfection.** NK cell line NK92 (ATCC: CRL-2407) was cultured in Alpha Minimum Essential medium (α-MEM) containing 2 mM L-glutamine, 1.5 g l$^{-1}$ sodium bicarbonate, 0.1 mM 2-mercaptoethanol, 0.02 mM folic acid, 500 U ml$^{-1}$ recombinant IL-2, 12.5% horse serum and 12.5% FBS. For NK92 transfection, cells ($1 \times 10^6$) were resuspended in 100 μl Nucleofector Solution buffer (Lonza) containing 5 μg DNA, followed by transfection using the Nucleofector Program Y-001 on Amaxa nucleofector II device (Lonza). Cells were recovered in α-MEM for 6 h, followed by flow cytometric sorting for viable cells and further culture[48]. For BM cell transfection, pSIN-EF2-IRES-EGFP lentiviral vectors carrying the indicated genes were transfected into HEK293T cells (maintained by our laboratory) together with psPAX2 and pMD2.G, followed by concentration through ultracentrifugation on 50,000 g for 2 h. Pellets of lentivirus were resuspended in serum-free α-MEM media. Donor BM cells were infected with lentiviruses by centrifugation at 500 g for 1.5 h in the presence of 8 μg ml$^{-1}$ Polybrene (Sigma-Aldrich) and then incubated at 37 °C for 18 h, followed by sorting for GFP$^+$ cells that were used for transplantation into lethally irradiated recipient mice[49].

**Animals.** Mouse experiments complied with ethical regulations and were approved by the Institutional Animal Care and Use Committees at the Institute of Biophysics, Chinese Academy of Sciences. Wash$^{flox/flox}$ mice were generated as previously described[27]. Arid1a$^{flox/flox}$ mice were provided by Dr. Ze-Guang Han (Chinese National Human Genome Center, Shanghai, China). Ncr1-Cre mice were generated by Beijing Biocytogen (Beijing, China). Rorc-Cre mice and RORγt-GFP mice were purchased from Jackson Laboratory. All mice were in a C57/BL6 background. Female mice at an age of 8 weeks were used in this study unless mentioned in the text.

**C. rodentium infection.** C. rodentium was a gift from Dr Baoxue Ge (Shanghai Institutes for Biological Sciences, Chinese Academy of Sciences). Mice were orally infected with C. rodentium ($2 \times 10^9$ per mouse), followed by survival rate and body change examination in the following days. Mice were sacrificed to examine colon pathology and bacterial loads on day 8 post infection.

**Intestinal lymphocyte separation and flow cytometry.** Intestinal lymphocytes were separated from intestines as described[1]. Briefly, intestines were cut into pieces with 0.5 cm length post removing fat tissues and Peyer's patches, followed by digestion in HBSS buffer containing 15 mM HEPES, 5 mM EDTA and 10% FBS. To obtain IELs, intestines were digested in the above buffer on a wheeler for 30 min at 37 °C for three times to collect the detached cells from the intestinal tissues. To obtain LPLs, the remaining intestinal tissues were further digested in HBSS buffer containing 10% FBS, 0.2 mg ml$^{-1}$ collagenase and 0.2 mg ml$^{-1}$ DNase I (Sigma-Aldrich). Detached cells were sifted through 50 μm cell strainers for further analysis. For flow cytometry, cells were stained with fluorescence-conjugated primary antibodies for 1 h on ice. Cells were then fixed and permeabilized, followed by nuclear staining of transcription factors. Cells were analysed or sorted by a cell sorter (BD AriaIII). Data were analysed using the FlowJo 7.6.1 software.

**BrdU incorporation and cell cycle analysis.** Mice were intraperitoneally injected with BrdU (7 mg kg$^{-1}$ body weight) for 16 h, followed by flow cytometry of BrdU signals in LT-HSCs. For staining BrdU in the nucleus, cells were fixed and permeabilized in fixation/permeabilization buffer (eBioscience) according to manufacturer's instructions. Permeabilized cells were stained with fluorophore-conjugated anti-BrdU antibody for 2 h, and washed with diluted permeabilization buffer for three times. For cell cycle analysis using Hoechst/Pyronin Y staining strategy, live cells were harvested from RORγt-GFP reporter mice and resuspended in 200 μl IMDM medium containing 10 μg ml$^{-1}$ Hoechst33342 for 45 min at 37 °C, followed by addition of Pyronin Y to a final concentration of 0.5 μg ml$^{-1}$ for 15 min at 37 °C.

**Histology.** Colons were fixed in 4% paraformaldehyde (PFA) (Sigma-Aldrich) for 12 h. Fixed tissues were washed for twice using 75% ethanol and embedded in paraffin, followed by sectioning and staining with haematoxylin and eosin according to standard procedures.

**Bone marrow transplantation.** Bone marrow transplantation (BMT) was performed as previously described[50]. Briefly, $2 \times 10^6$ BM cells were co-transplanted with $2 \times 10^6$ WT CD45.1 BM cells into lethally irradiated mice. For knockdown or overexpression, $10^7$ BM cells were infected with lentiviruses carrying shRNAs or overexpression sequences, followed by culture in HSC culture medium StemPro-34

supplemented with cytokines for 36 h. GFP positive LT-HSCs were sorted through flow cytometer, followed by BMT of $2 \times 10^6$ BM cells with $2 \times 10^6$ WT CD45.1 BM cells into lethally irradiated mice. CD45.2 and CD45.1 C57BL/6 mice were bred in SPF conditions. Mouse experiments complied with ethical regulations and were approved by the Institutional Animal Care and Use Committees at the Institute of Biophysics, Chinese Academy of Sciences.

**Immunofluorescence assay.** Immunostaining was performed as previously described[51]. Briefly, sorted cells were sticked on 0.01% poly-L-Lysine treated coverslips and fixed with 4% PFA for 10 min, followed by permeabilization with 0.5% Triton X-100 for 20 min at room temperature (RT). Primary antibodies were added for 2 h at RT post blocking with 10% donkey serum for 30 min. Samples were further stained with Alexa488-, Alexa594- or Alexa405-conjugated secondary antibodies, followed by confocal microscopy (Olympus FV1000).

**DNA FISH.** Cells were fixed with 4% PFA containing 10% acetic acid for 15 min at room temperature, followed by replacement with 70% ethanol at $-20$ °C. Cells were then incubated in buffer containing 100 mM Tris-HCl (pH 7.5), 150 mM NaCl, followed by cytoplasm digestion in 0.01% pepsin/0.01 N HCl for 3 min at 37 °C. Cells were further fixed in 3.7% PFA and replaced with ethanol to a final concentration of 100%. Cells were air dried and washed with $2 \times$ SSC, followed by blocking with buffer containing 100 mM Tris-HCl (pH 7.5), 150 mM NaCl, 0.05% Tween 20, 3% BSA for 20 min. Cells were then denatured in 70% formamide/$2 \times$ SSC, and incubated with fluorescence-labeled DNA probes overnight. Cells were counterstained with DAPI for nucleus post washing with PBS.

**Western blot.** Cells were lysed with RIPA buffer (150 mM NaCl, 0.5% sodium deoxycholate, 0.1% SDS, 1% NP40, 1 mM EDTA, 50 mM Tris (pH 8.0)), followed by separation with SDS-PAGE. Samples were then transferred to NC membrane and incubated with primary antibody in 5% BSA. After washing with TBST three times, membranes were incubated with HRP-conjugated secondary antibodies for visualization. $1 \times 10^5$ cells were used for a single sample for whole cell blots. See Supplementary Fig. 6 for uncropped blots.

**Co-immunoprecipitation assay.** For co-IP experiments, $4 \times 10^5$ NKp46$^+$ ILC3 cells (pooled from 20 RORγt-GFP reporter mice) were lysed in hypotonic buffer (10 mM HEPES, 1.5 mM MgCl$_2$, 10 mM KCl, 0.5 mM DTT) to remove cytoplasmic contents. Nuclear pellets were lysed in RIPA lysis buffer for 1 h, followed by incubation with anti-WASH antibody for 4 h post preclearance with protein A/G agarose. WASH containing complex was further precipitated by protein A/G agarose. Immunoprecipitates were washed with RIPA buffer, followed by immunoblotting with antibodies against WASH and Arid1a. $4 \times 10^5$ Arid1a deleted ILC3 cells sorted from $Arid1a^{flox/flox}Rorc$-Cre mice were also lysed and immunoprecipitated with anti-WASH antibody for co-IP assay as a control.

**Chromatin immunoprecipitation.** ChIP was performed using MAGnify ChIP kit with manufacturers' instructions. Briefly, cells sorted from intestines were cross-linked with 1% formaldehyde at 37 °C for 10 min, and then quenched with 0.125 M lysine, followed by swelling in lysis buffer (50 mM Hepes, pH 7.5, 140 mM NaCl, 1% Triton X-100, 0.1% NaDeoxycholate, and Protease Inhibitor Cocktail Set III) for 30 min on ice. Chromatin was sheared to a mean length of 400 bp by sonication. After being de-cross-linked by RNase, proteinase K, and heat, input genomic DNA was precipitated with ethanol and quantified in a GeneQuant 100 spectrophotometer (GE Healthcare). Chromatin was precleared with protein A/G-agarose, followed by incubation with indicated antibodies at 4 °C overnight and further incubation with protein A/G-agarose for 2 h. Beads were washed with washing buffer (10 mM Tris, pH 8.0, 250 mM LiCl, 0.5% NP-40, 0.5% Nadeoxycholate, 1 mM EDTA) three times and eluted with elution buffer (50 mM Tris, pH 8.0, 1% SDS, and 10 mM EDTA). Eluates were de-cross-linked by RNase, proteinase K, and heat, and DNA was extracted with phenolchloroform, followed by ethanol precipitation. For each ChIP experiment, $2 \times 10^4$ cells were used. 5% of nuclear extracts served as inputs. Immunoprecipitated DNAs were further analysed by real-time PCR. $2 \times 10^4$ NKp46$^+$ ILC3 cells were pooled from 4–10 WT or WASH knockout mice for each group.

**RNA interference.** RNA interference was performed according to pSUPER system instructions (Oligoengine). pSUPER vectors enclosing target sequences were constructed[52]. WASH target sequences were: #1, 5′-GCCAGAGCTAGAGAA TGAA-3′; #2, 5′-AGCGCAAACTGG AGAAGAA-3′.

**RT-PCR assay.** Total RNA was extracted from cells using Trizol reagent and cDNA was reverse-transcribed using Superscript II (Invitrogen). RT-PCR was performed using StarScript II Two-step RT-PCR Kit (Genestar) with the following primers: *Wash*, forward, 5′-CTCCTTGGCCCAGGCTAAG-3′, reverse, 5′-CTG CAGAGAGCCCGCTCATCCAG-3′; *Arid1a*, forward, 5′-ATCTTCGCAGCTGCTG ACTCC-3′, reverse, 5′-GGCATCCTGGATTC CGACTGAG-3′; *Ahr*, forward,

5′-AAGAAAGGGAAGGACGGAGC-3′, reverse, 5′-CTGCCC TTTGGCATCAC AAC-3′.

**Recombinant protein preparation.** cDNAs were cloned from a bone marrow cDNA library[48]. WASH or Arid1a was subcloned to pGEX-6 P-1 for GST-tagged protein expression or to pET-28a for His-tagged protein purification. Plasmids were transformed into *E. coli* strain BL21 (DE3). DE3 clones were cultured (OD$_{600}$ = 0.6), followed by induction with 0.2 mM IPTG at 16 °C for 24 h. Cells were collected and lysed by supersonic, followed by purification through Ni-NTA resins or GST resins. GST-tagged proteins were cleaved by PreScission protease to remove GST tags.

**Luciferase assay.** Luciferase reporter was constructed as described[51]. *Ahr* promoter region ($-2,000$ to 0) was subcloned into pGL3 vector. Luciferase reporter vectors were cotransfected with pRL-TK (as an internal control reporter vector) into macrophages by electroporation. Luciferase assays were performed with guidelines provided by the manufacturer (Promega). For NK92 transfection, cells ($1 \times 10^6$) were resuspended in 100 μl Nucleofector Solution buffer (Lonza) containing 5 μg DNA, followed by transfection using the Nucleofector Program Y-001 on Amaxa nucleofector II device (Lonza). Cells were recovered in RPMI1640 media containing 4 mM L-glutamine, 1.5 g l$^{-1}$ sodium bicarbonate, and 10% heat-inactivated fetal bovine serum (FBS) for 6 h, followed by flow cytometry sorting for viable cells.

**Nuclear run-on assay.** Sorted cells were harvested in buffer containing 150 mM KCl, 10 mM Tris-HCl, 4 mM MgOAc with pH 7.4, followed by centrifugation to collect cell pellets. Pellets were lysed in buffer containing 150 mM KCl, 10 mM Tris-HCl, 4 mM MgOAc, and 0.5% NP-40, followed by sucrose density gradient centrifugation to prepare crude nuclei. Crude nuclei were incubated with 10 mM ATP, CTP, GTP, BrUTP and RNase inhibitor at 28 °C for 5 min. RNAs were extracted using TRIzol reagent with manufacturer's guidelines, followed by DNA digestion with DNase I. RNA transcripts were immunoprecipitated with antibody against BrdU, followed by reverse transcription and RT-PCR analysis. $5 \times 10^4$ cells were used for a single sample.

**DNaseI accessibility assay.** Nuclei were purified from cells according to the manufacturer's protocol with Nuclei Isolating Kit (NUC101-1KT, Sigma-Aldrich). Then nuclei were resuspended with DNase I digestion buffer and treated with indicated units of DNase I (Sigma-Aldrich) at 37°C for 5 min. $2 \times$ DNase I stop buffer (20 mM Tris Ph 8.0, 4 mM EDTA, 2 mM EGTA) was added to stop reactions. DNA was extracted and examined by PCR.

**Yeast two-hybrid screening.** Yeast two-hybrid screening was performed as described[27]. Briefly, WASH was cloned into pGBKT7 vector (BD-WASH). Yeast AH109 cells were transfected with BD-WASH and plasmids containing a mouse BM cDNA library (Clontech/Takara) and then plated on selective SD medium. Selected clones were isolated and sequenced. X-α-gal assay was carried out following manufacturer's instructions.

**Statistical analysis.** Student's *t*-test was used as statistical analysis by using Microsoft Excel[53].

**Data availability.** The authors declare that the data supporting the findings of this study are available within the article and its supplementary information files.

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

## Acknowledgements

We thank Dr Liang Zhou (University of Florida) for his technical support and critical discussions, Junying Jia and Yan Teng for their technical support. This work was supported by the National Natural Science Foundation of China (31530093, 31429001, 91640203, 91419308, 81601361, 31570872, 31471386 and 31671531), the Strategic Priority Research Programs of the Chinese Academy of Sciences (XDB19030203 and XDA12020219). Youth Innovation Promotion Association of CAS to S.W. and the China Postdoctoral Science Foundation to P.X. (2015M571141).

## Author contributions

P.X., J.L. and S.W. designed and performed experiments, analysed data. P.X. wrote the paper; B.Y. analysed data; Y.D. and Z.X. performed some experiments; Z.-G.H. provided Arid1a flox mice and analysed data; Z.F. initiated the study, organized, designed and wrote the paper.

## Additional information

**Competing interests:** The authors declare no competing financial interests.

