## [Peer Review File · Nature Communications]

Reviewers' comments:

Reviewer #1 (Remarks to the Author):

In this paper, Xia et al show that WASH is essential for the expansion of a subset of ILC3, the NKp46+ ILC3, whereas it is dispensable for other ILC3 subsets as well as for ILC1 and ILC2. Mechanistically, they show that WASH is required for inducing the expression of AHR. Specifically, WASH is required to recruit Arid 1 and the BAF complex to the promoter of AHR, promoting AHR transcription.

This paper is novel and interesting. The impact of Arid1 and WASH on AHR expression is unprecedented and advances the field. The authors generate several conditional KO mice to prove their conclusions.

However, there are several weak points that the authors should address to increase the quality of the data presented and make this paper more convincing

1. The authors perform many reconstitution experiments in NKp46+ ILC3 derived from conditional WASH deficient or Arid-deficient mice. However, the number of NKp46+ ILC3 in these mice is extremely low, it is hard to believe that these few cells can be effectively transfected or transduced with DNA or RNA encoding Arid1, WASH or their mutants. The authors should provide details on how many cells are transduced, how many mice are used to recover a sufficient amount of cells for each experiment, how many times the experiment has been repeated. Moreover, the authors should specify the transduction efficiency of NKp46+ ILC3 and how they determine that.

2. The authors should also provide similar information for the ChIP experiments performed using WASH-deficient -NKp46+ ILC3.

3. ILC1 have been examined in the intestinal epithelium. However, it would be important to assess whether WASH has any impact on liver ILC1, which are more clearly defined in mouse. Moreover, when the authors analyze NKp46-Cre x WASH fl/fl mice, NK cells should be also examined

4. In bone marrow chimera experiments, the authors should show the degree of chimerism. It well known that ILC3 subsets may be partially radioresistant and may not be replaced by BM cells so easily. The authors should also clarify why they analyze BM chimeras at 16 weeks. Usually chimeras are examined earlier

5. The paper shows a defect of NKp46+ ILC3 in steady state. It would be important if the defect persists during infections, such as C. rodentium infection. It is also possible that during infection, the defect may extend to other ILC3 subsets

Minor point

In the introduction, the authors state that the development of ILC is largely unknown. This statement is incorrect, there are very good paper that have advanced the field and should not be underestimated.

Reviewer #2 (Remarks to the Author):

Xia et al. reported here that WASP is required for the maintenance of NKp46+ ILC3 cells through promoting AHR expression. The data are interesting. However, experiments need to be rigorously performed to support the conclusions.

Specific comments:

1) Fig. 1 and others: The authors have shown absolute numbers for most of the data. The percentages should be shown as well. Gating strategy for Fig. 1a-1c need to be discussed.

2) Fig. 1d: Has the Ncr1-Cre line described here been authenticated in the literature? If not, more controls need to be included to show the deletion specificity.

3) Fig. 1h: Are RORgt+ T cells in the gut, including Th17 and Th22 cells changed in Wash f/f Rorc-

cre mice?

4) Fig. 1e: The authors use Peyer's patches as readout for LTi cells. It is not accurate.

Cryptopatches should be examined for that purpose.

5) Fig. 3b: The authors stated that Ahr expression is modest in DN ILC3s and almost undetectable in CD4+ ILC3s. The data and the literature do not support this conclusion at all.

6) Fig. 3f: If WASH is expressed low in DN or CD4+ ILC3s as shown in Fig. S1a and WASH indeed promotes Ahr expression as claimed, it is unclear why WASH overexpression cannot augment Ahr transcription in DN ILC3s or CD4+ ILC3s? Does DN or CD4+ ILC3s lack other factors than WASH for Ahr expression?

7) Fig. 3g: It is unclear how the luciferase assay was performed. Presumably, pGL3-AHR is a luciferase reporter driven by AHR responsive elements? It is essential to show the protein expression of AHR in all luciferase assays.

8) Fig. 3j and 5c: H3K4me1 is a marker for enhancers especially poised enhancers, and not for the promoters. H3K4me3 should be used for the promoters.

9) Fig. 3k: It is unclear how the experiment was performed? Raw data need to be shown.

Repressive histone markers need to be examined. In addition, more accurate assay need to be performed to determine chromatin conformation of the Ahr promoter.

10) Fig. 4c: It is impressive to show the co-IP biochemical data in primary NKp46+ ILC3s. How many cells was this experiment performed with? More controls need to be included to show the specificity (e.g., not due to contamination of genomic DNA).

11) The author claimed that WASH recruits Arid1a to Ahr promoter to activate AHR expression.

The data do not directly support this statement. ChIP-ReCHIP needs to be done to carefully examine the interaction between WASH and Arid1a and their cooperative action at the promoter of AHR.

Point-by-point response to reviewers' comments

Reviewer #1

In this paper, Xia et al show that WASH is essential for the expansion of a subset of ILC3, the NKp46+ ILC3, whereas it is dispensable for other ILC3 subsets as well as for ILC1 and ILC2. Mechanistically, they show that WASH is required for inducing the expression of AHR. Specifically, WASH is required to recruit Arid 1 and the BAF complex to the promoter of AHR, promoting AHR transcription. This paper is novel and interesting. The impact of Arid1 and WASH on AHR expression is unprecedented and advances the field. The authors generate several conditional KO mice to prove their conclusions. However, there are several weak points that the authors should address to increase the quality of the data presented and make this paper more convincing

1. The authors perform many reconstitution experiments in NKp46+ ILC3 derived from conditional WASH deficient or Arid-deficient mice. However, the number of NKp46+ ILC3 in these mice is extremely low, it is hard to believe that these few cells can be effectively transfected or transduced with DNA or RNA encoding Arid1, WASH or their mutants. The authors should provide details on how many cells are transduced, how many mice are used to recover a sufficient amount of cells for each experiment, how many times the experiment has been repeated. Moreover, the authors should specify the transduction efficiency of NKp46+ ILC3 and how they determine that.

Answer: This is a good point. In our transfected or transduced experiments, we actually infected bone marrow cells with lentiviruses carrying indicated DNAs or RNAs, followed by bone marrow transplantation. We usually infected 1×10^7 bone marrow cells with lentiviruses and used 2×10^6 BM cells for engraftment. Infected BM cells (2×10^6) were co-transplanted with 2×10^6 WT CD45.1 BM cells into lethally irradiated mice. We then sorted donor-derived NKp46+ ILC3 cells for further analysis. We pooled NKp46+ ILC3 cells from 4-10 mice for each group and repeated these experiments for three times. We used lentiviruses with a GFP reporter to infect bone marrow cells and tested transfection efficiencies by analyzing GFP positive cells 36 h post infection. The transfection efficiency of BM cells with lentiviruses was over 20%. We provided these details in the methods section of our revised manuscript.

2. The authors should also provide similar information for the ChIP experiments performed using WASH-deficient -NKp46+ ILC3.

Answer: For ChIP experiments with NKp46+ ILC3s, we used a high efficient kit suitable for low cell amounts (MAGnify ChIP kit, Thermo Fisher). 2×10^4 NKp46+ ILC3 cells were pooled from 4-10 WT or WASH knockout mice, followed by crosslinking with 1% formaldehyde at 37°C for 10 min, then quenched with 0.125 M lysine. Cells were lysed and chromatin was sheared by sonication. Chromatin was then precleared with protein A/G-agarose, followed by incubation with indicated antibodies. Beads were washed and eluted with elution buffer. Eluates were de-crosslinked with RNase, proteinase K, and heat. DNA was extracted with phenolchloroform, followed by ethanol precipitation for further PCR

analysis. 5% of nuclear extracts served as inputs. We added the details in the methods section of our revised manuscript.

3. ILC1 have been examined in the intestinal epithelium. However, it would be important to assess whether WASH has any impact on liver ILC1, which are more clearly defined in mouse. Moreover, when the authors analyze NKp46-Cre x WASH fl/fl mice, NK cells should be also examined

Answer: This is a good suggestion. We examined cell numbers of liver ILC1s in WASH deficient mice. We observed that WASH deficiency did not significantly change cell numbers of liver ILC1s compared to WT littermate control mice (new Fig. S2a). We also checked the cell numbers of NKp46⁺ROR γ T⁺ cells in *Wash*^{flx/flx}*Ncr1*-Cre mice. We found that WASH knockout did not significantly affect the cell numbers of NK cells (new Fig. S2b).

4. In bone marrow chimera experiments, the authors should show the degree of chimerism. It well known that ILC3 subsets may be partially radioresistant and may not be replaced by BM cells so easily. The authors should also clarify why they analyze BM chimeras at 16 weeks. Usually chimeras are examined earlier

Answer: We provided the degrees of chimerism for bone marrow chimera experiments (new Fig. S2c, e). We performed these reconstitution experiments and analyzed chimeras at 8 or 16 weeks. For our assayed cell populations of reconstituted chimeras, we achieved almost similar results at these two time points. We provided new experimental data for chimerism assays at 8 weeks (new Fig. 2, new Fig. S2d-g, new Fig. 6d-f, new Fig. S3a-g, new Fig. S4f-h and new Fig. S5a-d).

5. The paper shows a defect of NKp46+ ILC3 in steady state. It would be important if the defect persists during infections, such as *C. rodentium* infection. It is also possible that during infection, the defect may extend to other ILC3 subsets

Answer: We actually infected WASH deficient mice with *C. rodentium*. However, we observed that WASH deficient mice were not susceptible to bacterial infection (new Fig. S1n). Moreover, WASH deficient mice showed similar body weights post *C. rodentium* infection to WT mice.

Minor point:

In the introduction, the authors state that the development of ILC is largely unknown. This statement is incorrect, there are very good paper that have advanced the field and should not be underestimated.

Answer: We changed our wording.

Reviewer #2

Xia et al. reported here that WASP is required for the maintenance of NKp46+ ILC3 cells through promoting AHR expression. The data are interesting. However, experiments need to be rigorously performed to support the conclusions.

Specific comments:

1) Fig. 1 and others: The authors have shown absolute numbers for most of the data. The percentages should be shown as well. Gating strategy for Fig. 1a-1c need to be discussed.

Answer: We provided cell percentages in respective figures (new Fig. S1a-g and j). ILC1 cells were gated out from Lin⁻CD45⁺RORγt⁻CD49a⁺NK1.1⁺NKp46⁺ SI IEL cells. ILC2 cells were gated out from Lin⁻CD127⁺ST2⁺Sca-1⁺KLRG⁺RORγt⁻ SI LPL cells and ILC3 were gated out from Lin⁻CD45⁺CD127⁺RORγt⁺ SI LPL cells. αLP cells were gated out from Lin⁻ cKit^{low} CD127⁺α4β7⁺ BM cells. ILCP cells were gated out from Lin⁻CD127⁺α4β7⁺ PLZF⁺ BM cells and CHILP cells were gated out from Lin⁻ CD127⁺α4β7⁺CD25⁻CD244⁺Id2⁺ BM cells. We provided these gating strategies for Fig. 1a-1c (new Fig. S1a and d-f).

2) Fig. 1d: Has the Ncr1-Cre line described here been authenticated in the literature? If not, more controls need to be included to show the deletion specificity.

Answer: We generated the Ncr1-Cre strain and had been described in our previous study (Wang S, *Nat Commun*, 2016). We cited this literature.

3) Fig. 1h: Are RORγt+ T cells in the gut, including Th17 and Th22 cells changed in Wash f/f Rorc-cre mice?

Answer: We examined these cells in the gut of *Wash^{flox/flox};Rorc-cre* mice. We observed that WASH deficiency did not affect the cell numbers of these cells in gut (new Fig. S1I).

4) Fig. 1e: The authors use Peyer's patches as readout for LTi cells. It is not accurate. Cryptopatches should be examined for that purpose.

Answer: We examined cryptopatches in the small intestine and large intestine of *Wash^{flox/flox}* and *Wash^{flox/flox}Rorc-Cre* mice. We found that WASH deficiency did not affect cell numbers of LTi cells in cryptopatches (new Fig. 1e).

5) Fig. 3b: The authors stated that Ahr expression is modest in DN ILC3s and almost undetectable in CD4+ ILC3s. The data and the literature do not support this conclusion at all.

Answer: We are sorry for this wrong wording. We changed it in our revised manuscript.

6) Fig. 3f: If WASH is expressed low in DN or CD4⁺ ILC3s as shown in Fig. S1a and WASH indeed promotes Ahr expression as claimed, it is unclear why WASH overexpression cannot augment Ahr transcription in DN ILC3s or CD4⁺ ILC3s? Does DN or CD4⁺ ILC3s lack other factors than WASH for Ahr expression?

Answer: This is a very good point. We found that the expression of *Ahr* was regulated by WASH and Arid1a in NKp46⁺ ILC3s. However, in NKp46⁻ ILC3s, WASH did not accumulate on the *Ahr* promoter (new Fig. 3d). In addition, WASH was associated with Arid1a only in NKp46⁺ ILC3s (new Fig. S3j). Actually, WASH overexpression could not augment Ahr transcription in DN ILC3s or CD4⁺ ILC3s, suggesting other factors than WASH may be required for Ahr expression in these cells, which needs to be further investigated.

7) Fig. 3g: It is unclear how the luciferase assay was performed. Presumably, pGL3-AHR is a luciferase reporter driven by AHR responsive elements? It is essential to show the protein expression of AHR in all luciferase assays.

Answer: This is the case. pGL3-AHR was surely a luciferase reporter driven by AHR responsive elements. We also examined the expression levels of endogenous AHR proteins in luciferase assays (new Fig. 3g and new Fig. 5e). We added more details in the methods section and provided the expression levels of endogenous AHR proteins for these luciferase assays.

8) Fig. 3j and 5c: H3K4me1 is a marker for enhancers especially poised enhancers, and not for the promoters. H3K4me3 should be used for the promoters.

Answer: This is a good suggestion. We repeated these experiments with antibody against H3K4me3 (new Fig. 3j and new Fig. 5c).

9) Fig. 3k: It is unclear how the experiment was performed? Raw data need to be shown. Repressive histone markers need to be examined. In addition, more accurate assays need to be performed to determine chromatin conformation of the Ahr promoter.

Answer: We isolated nuclei from NKp46⁺ ILC3 cells of *Wash*^{flox/flox} and *Wash*^{flox/flox} *Rorc*-Cre mice with the Nuclei isolating Kit (Sigma-Aldrich). Then nuclei were resuspended with DNase I digestion buffer and treated with indicated units of DNase I (Sigma, USA) at 37°C for 5 min. 2X DNase I stop buffer (20 mM Tris Ph 8.0, 4 mM EDTA, 2 mM EGTA) was added to stop reactions. DNA was extracted and tested by PCR. We provided raw data in the new Fig. 3k (bottom panel). We provided this method in the methods section of our revised manuscript.

We examined H3K27me3 levels on the *Ahr* promoter in NKp46⁺ ILC3 cells (new Fig. 3l). We also performed a nuclear run-on assay to verify the transcriptional activity of *Ahr* promoter post WASH deletion (new Fig. 3m). Our data indicate that WASH deletion

suppresses Ahr expression in NKp46⁺ ILC3 cells. We addressed this point in our revised text.

10) Fig. 4c: It is impressive to show the co-IP biochemical data in primary NKp46⁺ ILC3s. How many cells was this experiment performed with? More controls need to be included to show the specificity (e.g., not due to contamination of genomic DNA).

Answer: We pooled NKp46⁺ ILC3 cells (1X10⁵) from five ROR γ t-GFP reporter mice for each group. To exclude contamination of genomic DNA, we treated cell lysates with DNase I prior to immunoprecipitation (new Fig. 4c). With DNase I treatment, anti-WASH antibody still precipitated Arid1a, indicating the direct interaction of WASH and Arid1a in primary NKp46⁺ ILC3s.

11) The author claimed that WASH recruits Arid1a to Ahr promoter to activate AHR expression. The data do not directly support this statement. ChIP-ReCHIP needs to be done to carefully examine the interaction between WASH and Arid1a and their cooperative action at the promoter of AHR.

Answer: We performed ChIP-ReCHIP assays as suggested. We found that anti-Arid1a antibody could precipitate *Ahr* promoter in anti-WASH antibody precipitates (new Fig. S3h). Moreover, anti-WASH antibody could also precipitate *Ahr* promoter in anti-Arid1a antibody precipitates (new Fig. S3i). These data suggest that WASH and Arid1a indeed accumulated on the *Ahr* promoter. The interaction of WASH and Arid1a was defined in the Fig. 4 and Fig. 5f. We stated these results in our revised text.

Reviewers' comments:

Reviewer #1 (Remarks to the Author):

The authors have addressed my comments and the revised manuscript is considerably improved. Minor revisions are still needed

- 1) the paper contains several typos and writing should be improved for clarity
- 2) When describing bone marrow chimera experiments in the text, It should be made clear that BM cells are transduced and then used for BM chimeras

Reviewer #2 (Remarks to the Author):

Representative immunofluorescence figures for the cryptopatches should be shown. There is a discrepancy regarding the actual cre-deleter used in this experiment. The figure shows Ncr1-cre, but the rebuttal letter indicates Rorc-cre. If it is Ncr-cre as shown in the figure, it would be inaccurate to claim "WASH deficiency did not affect the number of cryptopatches" (page 5, line 124), since Ncr-cre would delete genes in NKp46+ cells but not in LTI-like cells that are major cell component of cryptopatches. Indeed, Rorc-cre should be used for this important statement.

My previous request regarding Ahr expression in ILC3s (point 5). The authors didn't change the statement. As shown in Figure 3b, Ahr expression seems high in NKp46+, but definitely detectable in both DN and CD4+ ILC3s (actually only 2-fold less than that in NKp46+ cells). The statement (page 7, line 183) thus is inaccurate. Ahr protein should be examined preferably.

Figure 4: Co-IP assay using 10^5 NKp46+ ILC3s could be technically challenging unless the antibody specificity is extremely high, the proteins exclusively interact with each other (i.e., all WASH and Arid1a within a cell are in the same protein complex), and/or the expression of these proteins is very high in any given cell. However, the data seems not to support these assumptions (Figure S3J: Only 6% of NKp46+ ILC3s have co-localization of these two proteins). It is questionable how to detect these events biochemically in just 6,000 cells. Typically, biochemical assay such as co-IP requires much more cells (10^6 arrange) to detect endogenous protein-protein association due to its efficiency. The controls need to be shown to assure the specificity of the signals detected by given antibodies.

My previous point 6 regarding the finding that over expression of WASH cannot augment Ahr transcription in DN or CD4+ ILC3s should be discussed in the text.

Point-by-point response to reviewers' comments

Reviewer #1

The authors have addressed my comments and the revised manuscript is considerably improved.

Minor revisions are still needed:

1) The paper contains several typos and writing should be improved for clarity

Answer: We carefully revised our whole paper and corrected typos.

2) When describing bone marrow chimera experiments in the text, it should be made clear that BM cells are transduced and then used for BM chimeras

Answer: We described these bone marrow chimera experiments in the according text and figure legends of our revised manuscript.

Reviewer #2

Representative immunofluorescence figures for the cryptopatches should be shown. There is a discrepancy regarding the actual cre-deleter used in this experiment. The figure shows *Ncr1-cre*, but the rebuttal letter indicates *Rorc-cre*. If it is *Ncr-cre* as shown in the figure, it would be inaccurate to claim "WASH deficiency did not affect the number of cryptopatches" (page 5, line 124), since *Ncr-cre* would delete genes in NKp46+ cells but not in LTI-like cells that are major cell component of cryptopatches. Indeed, *Rorc-cre* should be used for this important statement.

Answer: We are sorry to make this wrong description. In fact, WASH conditional KO mice were obtained by crossing *Wash^{flox/flox}* with *Rorc-Cre* mice for this experiment, followed by examination of cryptopatches in WASH deficient mice. As shown in the new Fig. 1h, we showed representative images of cryptopatches with anti-ROR γ t and anti-NKp46 staining in the small intestine. We also added the calculations of cryptopatch numbers in the new Fig. 1h.

My previous request regarding *Ahr* expression in ILC3s (point 5). The authors didn't change the statement. As shown in Figure 3b, *Ahr* expression seems high in NKp46+, but definitely detectable in both DN and CD4+ ILC3s (actually only 2-fold less than that in NKp46+ cells). The statement (page 7, line 183) thus is inaccurate. *Ahr* protein should be examined preferably.

Answer: We are sorry for this mistake. Right, this is the case. Similar protein expression levels of AHR in ILC3 subpopulations were further validated by immunoblotting (new Fig. S2h). We addressed this issue in our revised text.

Figure 4: Co-IP assay using 10^5 NKp46+ ILC3s could be technically challenging unless the antibody specificity is extremely high, the proteins exclusively interact with each other (i.e., all WASH and Arid1a within a cell are in the same protein complex), and/or the expression of these proteins is very high in any given cell. However, the data seems not to support these assumptions (Figure S3J: Only 6% of NKp46+ ILC3s have co-localization of these two proteins). It is questionable how to detect these events biochemically in just 6,000 cells. Typically, biochemical assay such as co-IP requires much more cells (10^6 arrange) to detect endogenous protein-protein association due to its efficiency. The controls need to be shown to assure the specificity of the signals detected by given antibodies.

Answer: We previously generated a highly sensitive antibody against WASH (Xia P, *EMBO J*, 2013). WASH protein signals can be detected with this antibody quite well for immunoblotting (5×10^4 cells). In addition, anti-Arid1a antibody (purchased from Cell Signaling Technology) we used was also quite sensitive, which can pick up Arid1a signals from cell lysates of 5×10^4 cells by immunoblotting. We found that Arid1a signals could not be picked up by anti-Arid1a antibody from anti-WASH precipitates derived from Arid1a deleted ILC3s lysates (new Fig. 4c). By contrast, Arid1a signals were able to be detectable in anti-WASH precipitates derived from WT ILC3s lysates (new Fig. 4c). These results validated the specificity of these two antibodies we used. For the Figure S3J, we only counted strongly merged signals of WASH and Arid1a with fluorescence staining in ILC3s for co-localization percentage calculations. Another possibility could be that anti-WASH and anti-Arid1a antibodies could not sensitively pick up their respective protein fluorescence signals well in NKp46⁺ ILC3s. These possibilities might cause an artificial lower co-localization percentage, which resulted in an inconsistent result with that by immunoblotting assays.

Surely, the expression levels of WASH and Arid1a were very high in NKp46⁺ ILC3s. Moreover, the antibody specificity of anti-WASH and anti-Arid1a antibodies was extremely high and sensitive. For immunoblotting assays of these two proteins in ILCs, we used 1×10^5 cells and worked very well. Whereas we used 4×10^5 cells (pooled from 20 ROR γ t-GFP reporter mice) for these co-immunoprecipitation assays. We were able to obtain good results for our co-IP assays (new Fig. 4c). We also confirmed the specificity of anti-WASH and anti-Arid1a antibodies we used. We are sorry to confuse these two cell numbers for WB and co-IP assays in our previous descriptions. We added the specificity controls and addressed this point in the revised text, as well as revised our wrong descriptions in according legends.

My previous point 6 regarding the finding that overexpression of WASH cannot augment Ahr transcription in DN or CD4+ ILC3s should be discussed in the text.

Answer: We addressed this issue in our revised text

REVIEWERS' COMMENTS:

Reviewer #2 (Remarks to the Author):

My previous concerns have been appropriately addressed. The paper now has been improved significantly.

Point-by-point response

Reviewer #2:

My previous concerns have been appropriately addressed. The paper now has been improved significantly.

Answer: We thank the reviewer for careful review.